# Repeat-associated non-AUG translation in C9orf72-ALS/FTD is driven by neuronal excitation and stress

Thomas Westergard, Kevin McAvoy, Katelyn Russell, Xinmei Wen, Yu Pang, Brandie Morris, Piera Pasinelli, Davide Trotti[*] (iD) & Aaron Haeusler[**] (iD)

## Abstract

**Nucleotide repeat expansions (NREs) are prevalent mutations in a multitude of neurodegenerative diseases. Repeat-associated non-AUG (RAN) translation of these repeat regions produces mono or dipeptides that contribute to the pathogenesis of these diseases. However, the mechanisms and drivers of RAN translation are not well understood. Here we analyzed whether different cellular stressors promote RAN translation of dipeptide repeats (DPRs) associated with the G4C2 hexanucleotide expansions in C9orf72, the most common genetic cause of amyotrophic lateral sclerosis (ALS) and frontotemporal dementia (FTD). We found that activating glutamate receptors or optogenetically increasing neuronal activity by repetitive trains of depolarization induced DPR formation in primary cortical neurons and patient derived spinal motor neurons. Increases in the integrated stress response (ISR) were concomitant with increased RAN translation of DPRs, both in neurons and different cell lines. Targeting phosphorylated-PERK and the phosphorylated-eif2α complex reduces DPR levels revealing a potential therapeutic strategy to attenuate DPR-dependent disease pathogenesis in NRE-linked diseases.**

**Keywords** ALS; C9orf72; DPR; excitotoxicity; FTD
**Subject Categories** Genetics, Gene Therapy & Genetic Disease; Neuroscience

## Introduction

A nucleotide repeat expansion (NRE) mutation was recently identified in a non-coding region of the *C9orf72* gene (DeJesus-Hernandez *et al*, 2011; Renton *et al*, 2011). This *C9orf72* NRE mutation is a hexanucleotide repeat, $GGGGCC_n$ $(G_4C_2)_n$, and is currently the most prevalent genetic association for amyotrophic lateral sclerosis (ALS) and frontal temporal dementia (FTD) (Majounie *et al*, 2012b; Rademakers, 2012). Furthermore, the *C9orf72* NRE mutation is found in patients that have symptoms associated with a number of other neurological disorders such as Alzheimer's disease (Majounie *et al*, 2012a; Rollinson *et al*, 2012; Kohli *et al*, 2013), schizophrenia (Galimberti *et al*, 2014), progressive myoclonic epilepsies (van den Ameele *et al*, 2018), and as a Huntington's disease phenocopy (Hensman Moss *et al*, 2014). Symptomatic patients carrying the *C9orf72* NRE mutation have expansions that are 100s–1,000s of repeats in length, while non-affected individuals typically carry $\leq 30$ (van Blitterswijk *et al*, 2013; Suh *et al*, 2015). The *C9orf72* NRE mutation shares pathophysiological features described for many other NRE-linked neurological and neuromuscular disorders, which include somatic NRE-linked genome instability (DeJesus-Hernandez *et al*, 2011; Renton *et al*, 2011); the bidirectional transcription of the NRE locus generating both sense, $(G_4C_2)_n$, and antisense, $(C_4G_2)_n$, transcripts (Mori *et al*, 2013a; Zu *et al*, 2013); the formation of RNA foci (DeJesus-Hernandez *et al*, 2011); and the production of polypeptides through the unconventional non-AUG-dependent translation of the NRE region, frequently referred to as repeat-associated non-AUG-initiated (RAN) translation (Zu *et al*, 2011, 2013; Ash *et al*, 2013; Mori *et al*, 2013a).

Currently, three pathogenic mechanisms have been proposed to explain neurodegeneration in patients carrying the *C9orf72* NRE mutation: (i) Reduced sense transcript levels lead to haploinsufficiency that results in altered lysosomal dynamics and/or compromised inflammatory response (O'Rourke *et al*, 2016; Ji *et al*, 2017; Shi *et al*, 2018); (ii) the bidirectionally transcribed NRE region can form sense and antisense RNA foci and sequester essential ribonucleoproteins (Donnelly *et al*, 2013; Lee *et al*, 2013; Mori *et al*, 2013b; Haeusler *et al*, 2014; Zhang *et al*, 2015); and (iii) the repeat-containing RNA can undergo non-AUG-dependent translation. The unconventional translation of $(G_4C_2)_n$ has been shown to produce dipeptide repeats (DPRs) in all open reading frames of the *C9orf72* NRE mutation in patients (Zu *et al*, 2013). Recent mechanistic exploration of this translational phenomenon for the $(G_4C_2)_n$ sense transcript has suggested that the non-AUG-dependent translation initiates predominantly from a near-cognate AUG codon, near-cognate CUG codon, upstream of the repeat (Todd *et al*, 2013; Kearse *et al*, 2016; Tabet *et al*, 2018), and this non-AUG-dependent usage increases under specific cellular stress triggers in a cap-dependent and/or cap-independent manner (Green *et al*, 2017; Cheng *et al*, 2018; Sonobe *et al*, 2018). Ribosomal frameshifting, $\pm 1$ or 2 base

Department of Neuroscience, Jefferson Weinberg ALS Center, Vickie and Jack Farber Institute for Neuroscience, Jefferson University, Philadelphia, PA, USA
*Corresponding author. Tel: +1 215 955 8416; E-mail: davide.trotti@Jefferson.edu
**Corresponding author. Tel: +1 215 955 8630; E-mail: aaron.haeusler@Jefferson.edu

pairs, immediately upstream of the NRE, leads to the translation of all ORFs with varying levels of translational efficiency (Todd *et al*, 2013; Kearse *et al*, 2016; Green *et al*, 2017; Cheng *et al*, 2018; Tabet *et al*, 2018). This results in the translation of three unique DPRs from the sense transcript, $(G-A)_n$, $(G-P)_n$, and $(G-R)_n$. Additionally, the bidirectionally transcribed antisense transcript produces the DPRs, $(P-A)_n$, $(P-G)_n$, and $(P-R)_n$ and may share similar non-AUG-dependent translation mechanistic origins (Mori *et al*, 2013a; Zu *et al*, 2013).

The five unique DPRs generated from the bidirectionally transcribed *C9orf72* NRE have different cellular localizations (Ash *et al*, 2013; Zu *et al*, 2013; Wen *et al*, 2014), diverse protein interactomes (May *et al*, 2014; Lee *et al*, 2016), and show distinct physical properties *in vitro* (Freibaum & Taylor, 2017). In cell cultures, *Drosophila*, and mouse disease models, severe toxicity has been demonstrated by DPRs, particularly the arginine-rich species (Mizielinska *et al*, 2013; Kwon *et al*, 2014; Wen *et al*, 2014). In *C9orf72* patients, GP levels correlate with C9 FTD scores and GR levels correlate with degeneration in motor regions of C9 ALS patients (Gendron *et al*, 2017; Saberi *et al*, 2018). Therefore, it is important to understand the upstream mechanisms involved in triggering and initiating non-AUG translation to identify promising upstream therapeutic opportunities that could treat a number of NRE-linked neurological and neuromuscular disorders.

DPR pathology is primarily observed in neurons of *C9orf72* post-mortem patient tissues (Zu *et al*, 2013; Schludi *et al*, 2015), suggesting neuron-specific activation of mechanisms involved in non-AUG translation. Age-dependent hyperexcitability or hypoexcitability and increased vulnerability to excitotoxicity in neurons have been reported in *C9orf72* NRE models, including patient-derived motor neurons (Donnelly *et al*, 2013; Sareen *et al*, 2013; Wainger *et al*, 2014; Devlin *et al*, 2015; Shi *et al*, 2018) and patient tissue (Wainger & Cudkowicz, 2015). The downstream signaling that results from these events might potentially facilitate or activate mechanisms that drive non-AUG translation in neurons and explain the predominant neuronal localization of DPRs in patients. One potential outcome from these events is the dynamic translational regulation of phosphorylated α-subunit of eukaryotic translation initiation factor 2 (p-eif2α). P-eif2α-dependent translational regulation is a key response element to cellular stress that has been shown to increase non-canonical translation of mRNAs (Chesnokova *et al*, 2017) and is implicated to have a role in *C9orf72* NRE-linked RAN translation (Green *et al*, 2017; Cheng *et al*, 2018). In neurons, p-eif2α also allows for rapid activity-dependent alterations of proteins at the synapse, which is important for synaptic plasticity (Costa-Mattioli *et al*, 2009; Bellato & Hajj, 2016).

To further our understanding of the molecular and cellular mechanisms that drive non-AUG translation of the *C9orf72* repeat expansion, we developed a non-AUG-translated $(G_4C_2)_{188}$ *C9orf72* NRE Dendra2 fluorescent reporter construct to monitor non-AUG translation in various cell types, including primary neurons. Our results show that initiation of cellular stress yields a concomitantly increased production of DPRs via non-AUG translation and that these DPRs have long half-lives. In cortical and spinal motor neurons (sMNs), we demonstrate excitotoxic stress and repetitive neuronal activity act as promoters of non-AUG translation leading to increased DPR levels. We found that many of the cellular stressors or excitotoxic stressors ultimately converge on the integrated

stress response (ISR) and thereby lead to increased phosphorylation of eif2α in concert with increased non-AUG-dependent translation. Utilizing FDA-approved drugs that target the phosphorylation of eif2α or inhibiting effects of p-eif2α in the ISR, we were able to reduce non-AUG translation of DPRs. Together, these results provide new understanding of cell- and neuron-specific molecular triggers of non-AUG translation.

# Results

## *C9orf72* NRE undergoes non-AUG-dependent translation and produces long-lived DPRs that recapitulate DPR model features

To investigate possible pathogenic mechanisms that drive non-AUG-dependent translation of *C9orf72*-NRE, we first developed a *C9orf72* NRE fluorescent reporter construct. This construct contains the endogenous *C9orf72* 5′ gene region that extends from the beginning of exon 1A to the NRE mutation site (Fig 1A). Immediately downstream of the NRE mutation insertion site is a photoconvertible protein, Dendra2, with a C-terminal human influenza hemagglutinin (HA) affinity tag. The Dendra2-HA fusion protein lacks an AUG translation initiation codon, but is in the ORF that corresponds to the (GA) DPR when translating $(G_4C_2)_{\sim 188}$ placed in the NRE insertion site out of the sense strand. To fluorescently monitor the other two $(G_4C_2)_{\sim 188}$ DPR ORFs, the Dendra2-HA was shifted +1 or +2 base pairs relative to the NRE insertion site, thus generating a total of three different constructs that express a C-terminal fluorescent-tagged DPR from the *C9orf72* NRE $(G_4C_2)_{\sim 188}$ in ORF 1, 2, and 3, which correspond to GA, GP, and GR, respectively (Fig 1A). Because the *C9orf72* NRE could be bidirectionally transcribed (Mori *et al*, 2013a; Zu *et al*, 2013), we also inserted the antisense *C9orf72* NRE transcripts, $(C_4G_2)_{\sim 188}$, into the NRE insertion site to allow for RAN translation of the DPRs from the *C9orf72* NRE antisense strand in ORF 1, 2, and 3 that correspond to PA, PG, and PR, respectively (Fig EV1A). When transfected in cells, these constructs allow us to systematically monitor translated DPRs, collectively referred to hereafter as C9-DPR reporters, by quantifying the fluorescent intensity of the reporter protein tag Dendra2-HA (Figs 1 and EV1).

We began by characterizing non-AUG-dependent translation of the sense and antisense *C9orf72* NRE DPR reporter system. To accomplish this with appreciable DPR levels translated in the absence of an AUG (Green *et al*, 2017), we transiently transfected each of the six C9-DPR reporter constructs into HEK293T cells, then performed filter-trap binding assays probing for the c-terminus HA affinity tag of the DPR reporter or directly detecting the DPRs using DPR-specific antibodies. The results of this immunoassay demonstrated that the HA tag could be detected using the C9 DPR reporter in all open reading frames in the absence of an AUG codon and that there were measurable differences in sense DPR levels with the GA ORF predominantly the highest (Fig 1B). We also verified that each of these DPR reporter constructs was indeed generating all three N-terminus DPRs using DPR-specific antibodies. Dot blot analysis confirmed that one sense C9 DPR reporter construct transfected in HEK293T cells was indeed generating N-terminal DPRs in each of the three ORFs, although only one of these ORFs would be in frame with the Dendra2-HA tag for each construct (Fig EV1F). Additionally, immunofluorescence analysis demonstrated that the

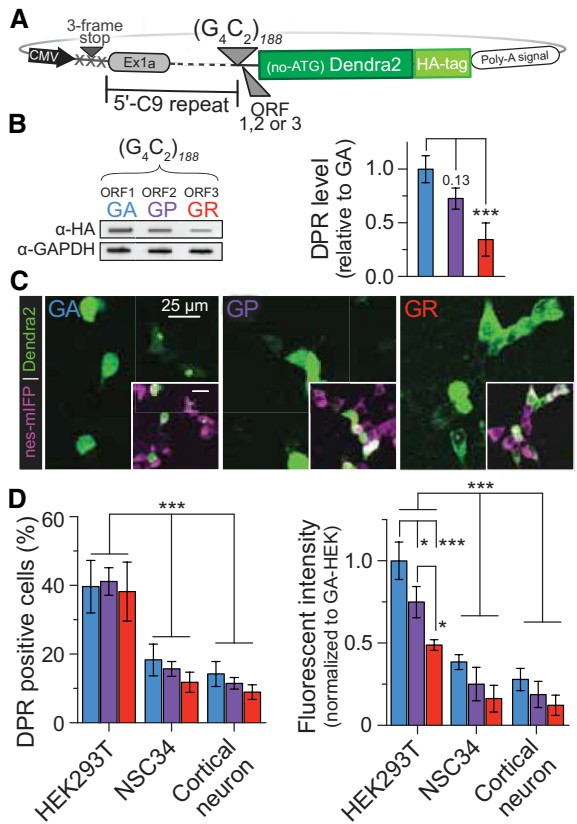

**Figure 1. *C9orf72*-based constructs containing the sense, $(G_4C_2)_{-188}$, NRE undergo pathologically relevant non-AUG-dependent translation with DPR levels that vary by cell type.**

A  Plasmid reporter constructs developed to monitor non-AUG-dependent translation using the photoconvertible fluorescent protein, Dendra2. The endogenous *C9orf72* 5′ gene region is immediately upstream, and Dendra2 is immediately downstream of the NRE insertion site. Each construct is modified +1 or +2 base pairs to fluorescently monitor all individual ORFs that are in-frame with the Dendra2 with an N-terminal HA affinity tag.

B  All ORFs for the sense non-AUG-dependent translation of the $(G_4C_2)_{n-188}$, DPR coding for (GA), (GP), (GR) can be detected and quantified using a filter-trap binding assay probing for the c-terminal HA tag of the C9 DPR reporter construct. The (GA) ORF is the most abundantly detected DPR and GR the lowest in HEK293T cell 24 h post-transient transfection of the C9 DPR reporter construct. $n = 5$. Error bars represent the mean $\pm$ SEM.

C  The *C9orf72* NRE non-AUG-dependent reporter constructs recapitulate localization patterns and pathological features identified in *C9orf72* NRE models. The Dendra2 (green) c-terminal fusion was used to monitor cellular localization and fluorescence intensity for each DPR translated from the $G_4C_2$ strand in HEK293T cells transiently cotransfected with an NES-mIFP-mIFP-NES (NES-mIFP) fusion protein cytoplasmic marker.

D  C9 DPR reporter constructs show cell-type variability in the number of cells that are DPR positive and the overall DPR levels. Transient cotransfections of HEK293T cells robustly undergo $G_4C_2$-Dendra2 non-AUG-dependent translation compared to NSC34 or rat primary cortical neurons. DPR-positive cells and DPR fluorescent intensity were analyzed and calculated 24 h post-transfection only in cells expressing the cotransfected NES-mIFP construct. DPR fluorescent intensity was normalized to NES-mIFP fluorescent intensity to adjust for cell-specific AUG-dependent translation levels; represented as fold-change relative to HEK293T. HEK293T: $n = 5$ with $m > 500$ cells analyzed per $n$. NSC34: $n = 5$ with $m > 500$ cells per $n$. Rat primary cortical neuron: $n = 5$ with $m > 40$ cells per $n$. Error bars represent the mean $\pm$ SEM.

Data information: ****P* < 0.001, **P* < 0.05. See also Figs EV1 and EV2. Source data are available online for this figure. Statistical comparisons were performed using a *t*-test.

Dendra2 tag colocalized with DPR staining in the corresponding frame (Fig EV1F). Consistent with the observations that only one frame is being monitored per construct, the localization patterns of the C9 DPR reporter are dependent on the ORF and give similar profiles as previously shown in transiently transfected AUG-initiated DPR fluorescent reporter systems (Wen *et al*, 2014; Figs 1C and EV1C). Specifically, in HEK293T, NSC34, and rat primary cortical neuronal cells, cotransfected with a NES-mIFP-mIFP-NES (NES-mIFP) cytosolic marker (Fig EV1E), we observed that the sense/antisense corresponding GA/PA is primarily cytoplasmic with GA being diffuse and punctate and PA diffuse, the GP/PG is more diffuse and ubiquitously distributed throughout the cell, and the GR/PR is punctate and predominantly nuclear at 24 h post-transfection (Figs 1C, and EV1C, EV2 and EV3). Additionally, comparisons among different cell models reveal potentially cell type-dependent differences in RAN translation efficiency independent of the C9 DPR reporter transcript levels (Figs 1D and EV1G) as shown with other NREs (Zu *et al*, 2011). For example, in HEK293T cells labeled by the cotransfected NES-mIFP (HEK293T[NES-mIFP]), the percentage of DPR-expressing cells is almost doubled compared to NES-mIFP-positive NSC34 (NSC34[NES-mIFP]) or rat primary cortical neurons (CN[NES-mIFP]). Furthermore, HEK293T[NES-mIFP] cells display higher relative fluorescent intensity for all DPRs in contrast to either NSC34[NES-mIFP] or rat CN[NES-mIFP] (Figs 1D, and EV1D and EV3). Together, these results validate that the C9 DPR reporter generates non-AUG-dependent translation products, which have localization patterns consistent with their AUG-initiated counterparts and with *C9orf72* NRE DPR pathological features frequently observed in patient-derived iPS neurons (Ash *et al*, 2013; Zu *et al*, 2013; Wen *et al*, 2014), but the efficiency of RAN translation may vary among cellular models.

In patients with the *C9orf72* NRE, there is an accumulation of specific DPRs in tissue, but little is known whether this could be in part due to intrinsic DPR turnover rates. The photoconvertible Dendra2 protein has proven to be a valuable tool that can be used to measure protein turnover while having minimal influence on the turnover rate kinetics of the fused protein (Hamer *et al*, 2010; Sattar-zadeh *et al*, 2015). Therefore, we cotransfected C9 DPR reporters with NES-mIFP in HEK293T cells, photoconverted the Dendra2 from fluorescent green to red (Fig EV3), and then monitored the decay of red fluorescent intensity longitudinally for sense and antisense C9 DPRs (Fig EV3). Sense and antisense DPRs have long predicted half-lives (> 100 h), with most DPRs half-lives calculated to be > 200 h (Fig EV3, Table EV1). These measurements also indicate that the increased granularity, or more punctate expression profile, of the DPR correlates with an increase in the predicted half-life, as measured by the differences between the fluorescent decay kinetics of diffuse smaller GA granules compared with larger punctate GA granules (Fig EV3 and Table EV1). Additionally, we identified that the cellular compartmentalization of the DPR dictates its kinetics of decay. Specifically, using the NES-mIFP live-cell-imaging cytosolic marker we measured an increase in the predicted half-life for GR when it localizes in the nucleus compared to the cytoplasm. In general, however, all six DPR species have relatively long predicted half-lives that are dependent on the granularity and/or cellular localization, and therefore, the accumulation of DPRs in a short timescale is most likely due to increased DPR production rather than decreased turnover rates.

**Non-AUG-dependent translation of the *C9orf72* NRE is increased by stress-inducing agents and neuronal excitotoxic stress**

To better understand cellular pathways that drive non-AUG-dependent translation, we examined the effect of activating a variety of unique or overlapping molecular pathways on the non-AUG-dependent translation of DPRs, using the previously employed cotransfection approach (as in Fig 1D). Specifically, unconventional translation mechanisms have been shown to be utilized under conditions of cellular stress. Therefore, we employed a panel of compounds that increased cellular stress through pathways such as ER stress, oxidative stress, or excitotoxic stress (Table EV2). Using only the sense C9 DPR reporter constructs, which carry the endogenous 5′ repeat flanking region from the *C9orf72* gene, we first performed dose–response experiments by treating NSC34 cells with a concentration range of a panel of known stress-inducing compounds (Figs 2A and EV4, Table EV2). NSC34 give the advantage of high transfection efficiency and low basal non-AUG translation, which together increase the ability to measure significant shifts in non-AUG-driven DPR levels. All of the compounds tested caused a dose-dependent increase in the percentage of C9 DPR reporter expressing NSC34$^{NES-mIFP}$ cells (Figs 2A and EV4). In Table EV2, we present the calculated $EC_{50}$ of the different compounds. We then confirmed that treating the cotransfected NSC34$^{NES-mIFP}$ cells with the calculated $EC_{50}$ stressor concentrations indeed produced translated DPRs by directly measuring DPR levels using a filter-trap binding assays and by measuring DPR fluorescent intensity levels in NSC34$^{NES-mIFP}$ cells. All stress-inducing compounds lead to a densitometrically quantifiable increase for all three sense C9 DPR protein levels upon treatment with a single stressor dose (Fig 2B). Furthermore, this relative increase in DPR levels is directly comparable to the DPR fluorescent intensity that was measured in stressor-treated NSC34$^{NES-mIFP}$ cells (Fig 2C). Increases in DPR protein levels were independent of transcript levels of the DPR reporter construct (Fig EV1G). Taken together, these results confirm that measuring the number of C9 DPR reporter-positive cells, the DPR fluorescent intensity, or DPR protein levels using DPR-specific antibodies generates similar results that ubiquitously demonstrate stress-inducing compounds increase non-AUG-dependent translation of the *C9orf72* NRE.

The increase in *C9orf72* NRE non-AUG-dependent translation in response to stress-inducing compounds might also depend on the cell type in which the stress is elicited. To investigate this hypothesis, we compared the extent of stress-inducing responses in C9 DPR RAN translation in NSC34 $^{NES-mIFP}$ cells with the response to stressors elicited in rat CN$^{NES-mIFP}$ and *C9orf72* NRE patient-derived iPS spinal motor neurons (sMNs). NSC34$^{NES-mIFP}$ cells, rat CN$^{NES-mIFP}$, and *C9orf72* NRE mutation patient-derived iPS sMNs treated with stress-inducing compounds show comparable results with respect to increases in C9 DPR reporter fluorescent intensity or increased DPR immunofluorescent intensity as seen for the iPS sMNs (Fig 2D). In contrast, both cortical and iPS-derived sMNs displayed a more robust increase in DPR levels in the presence of the excitotoxic stressors, glutamate and homocysteine (HC), compared to NSC34$^{NES-mIFP}$ cells which demonstrated insignificant densitometric DPR levels and less significant fluorescent DPR levels in the presence of glutamate. This observation may be consistent with evidence showing that NSC34 cells have decreased calcium permeability responsiveness to excitotoxic stress even when differentiated into neuronal-like cells (Madji Hounoum *et al*, 2016). Taken together, these results demonstrate that a spectrum of compounds, which activate a number of cellular stress pathways, all increase non-AUG-dependent translation with neurons showing heightened sensitivity to excitotoxic stressors.

**Glutamate-induced increase in *C9orf72* NRE non-AUG-dependent translation is facilitated by ionotropic glutamate receptors**

We examined the role of glutamate receptors in increasing non-AUG-dependent translation following excitotoxic stress by measuring DPR levels in the presence of glutamate receptor agonists and/or antagonists. Previous work has shown that neurons derived from *C9orf72* NRE patient iPS cells show increased glutamate receptors (Shi *et al*, 2018), age-dependent glutamate sensitivity, and hyperactivity/hypoactivity (Donnelly *et al*, 2013; Sareen *et al*, 2013; Wainger *et al*, 2014; Devlin *et al*, 2015), and motor neuron excitotoxicity may be a primary mechanism that contributes to ALS disease pathogenesis (Starr & Sattler, 2018). Furthermore, HC levels increase moderately with age in the central nervous system, and

---

**Figure 2. Cellular stress or neuronal excitotoxic stress increases DPR levels for all *C9orf72* NRE ORFs in rat cortical neurons and in patient-derived iPS spinal motor neurons.**

A  Quantitative fluorescent microscopy imaging shows a dose-dependent increase in the number of C9 DPR reporter-positive cells with increasing concentrations of an apoptotic stimulator, staurosporine (SSP). The C9 DPR reporter in the GP frame was utilized for these experiments. *n* = 4.

B  Filter-trap binding assays using DPR-specific antibodies show all DPR levels are increased for the sense-coding C9 DPR reporter when NSC34 cells are challenged with stress-inducing compounds. These stress-inducing compounds broadly cover distinct or overlapping cellular pathways (see Table EV2). Antibodies specific for the *C9orf72* NRE DPRs, poly-GA, -GP, or -GR, were used to quantify DPR levels normalized to GAPDH 24 h post-transient transfection of the C9 DPR reporter constructs. *n* = 4.

C  Quantitative fluorescent microscopy imaging shows that NSC34, rat primary cortical neurons, have increased relative DPR fluorescent intensity levels following compound induced cellular stress. In NES-mIFP cotransfection-positive NSC34 and rat primary cortical neurons, there is a significant increase in C9 DPR reporter fluorescent intensity levels 24 post-treatment with most stress-inducing compounds. Fluorescent intensity was not normalized to NES-mIFP fluorescent levels. NSC34: *n* = 10 with *m* > 500 cells analyzed per *n*. Cortical neuron: *n* = 6 with *m* > 40 cells per *n*.

D  Endogenously expression of DPRs in iPS spinal motor neurons (iPS sMN) derived from *C9orf72* NRE patients increases following induction of cellular stress. DPR fluorescent intensity in iPS sMN was measured using the DPR-specific antibodies described above. IPS sMN: *n* = 4 with *m* > 20 cells per *n*. Statistical comparisons were performed using an uncorrected Fisher's exact test.

Data information: All changes in DPR levels are shown relative to non-stressed or DMSO-only (vehicle) treated CTRL cells unless otherwise noted. Control (CTRL), thapsigargin (TG), menadione (Mena), staurosporine (SSP), diamide (Daim), cytochalasin D (Cyto D), etoposide (Etop), leptomycin B (Lepto), homocystine (HC), sodium arsenite (NaArs), tunicamycin (TM), glutamate (Glut). Statistical comparisons in (A) were calculated using *t*-tests per concentration, and (B) and (C) were calculated using an uncorrected Fisher LSD two-way ANOVA for CTRL versus treatment (****$P$ < 0.0001, ***$P$ < 0.001, **$P$ < 0.01, *$P$ < 0.05). See also Table EV2 and Figs EV2 and EV3. Source data are available online for this figure. Error bars represent the mean ± SEM.

significantly elevated HC levels correlate with multiple neurological disorders, such as age-related dementias and ALS (Ho et al, 2002; Obeid & Herrmann, 2006; Zoccolella et al, 2010). Therefore, to further characterize the effect of non-AUG-dependent translation by excitotoxic stressors, we tested the effect of different glutamate receptor agonists in rat CN[NES-mIFP]. All agonists cause a significant increase in non-AUG-dependent translation compared to untreated control, as measured by the quantitative fluorescent intensity of the

C9orf72 DPR reporter in CN[NES-mIFP] (Fig 3A). Specifically, glutamate, HC, and NMDA were significantly more potent inducers of non-AUG-dependent than AMPA, which moderately increased DPR levels. Pretreatment of primary cortical neurons with specific antagonists prior to agonist treatment reverses the excitotoxic stressor-induced non-AUG-dependent translation. In particular, inhibiting the NMDA receptor complex using MK801 shows the most dramatic reversal of excitotoxic stress-induced non-AUG-dependent

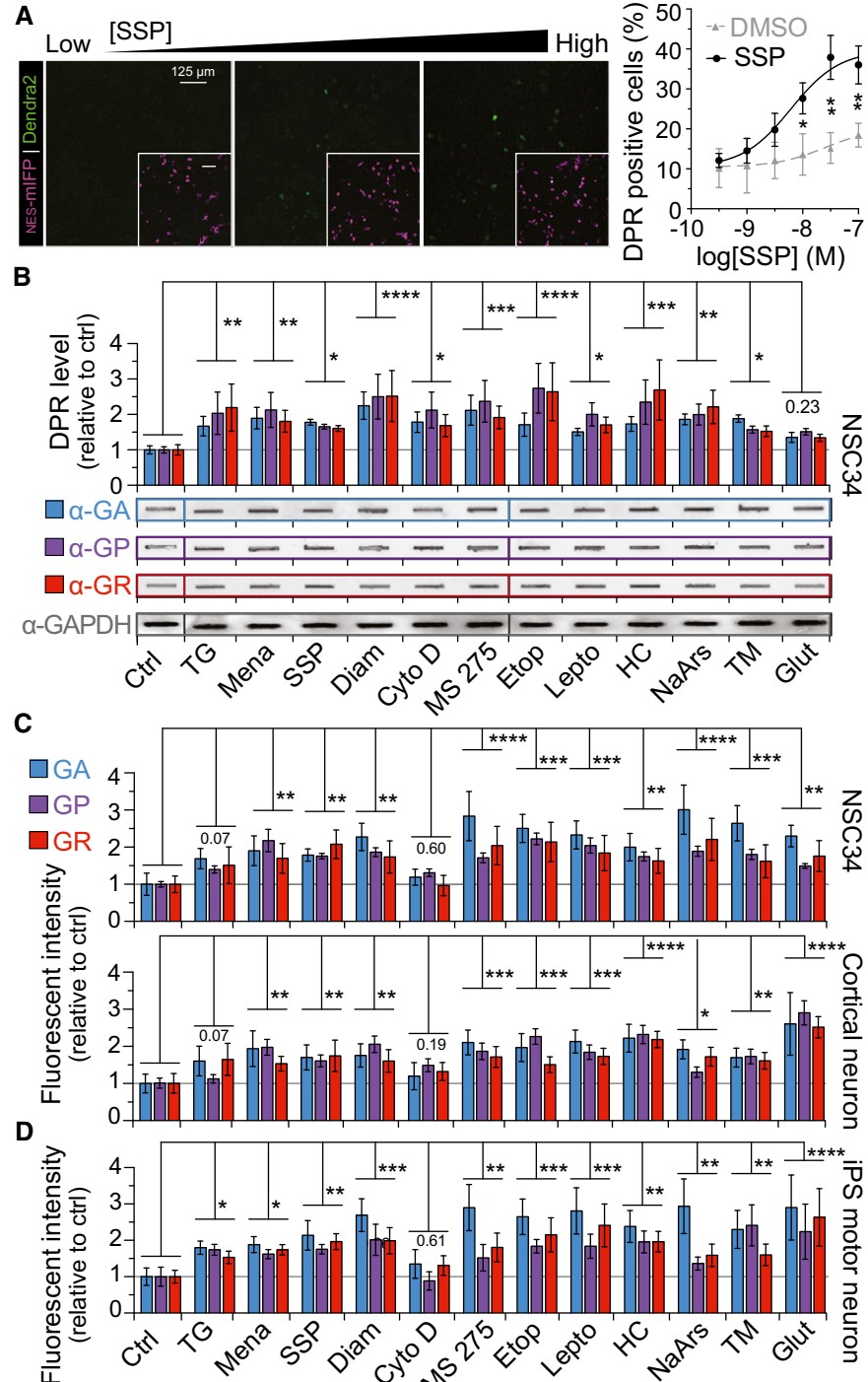

Figure 2.

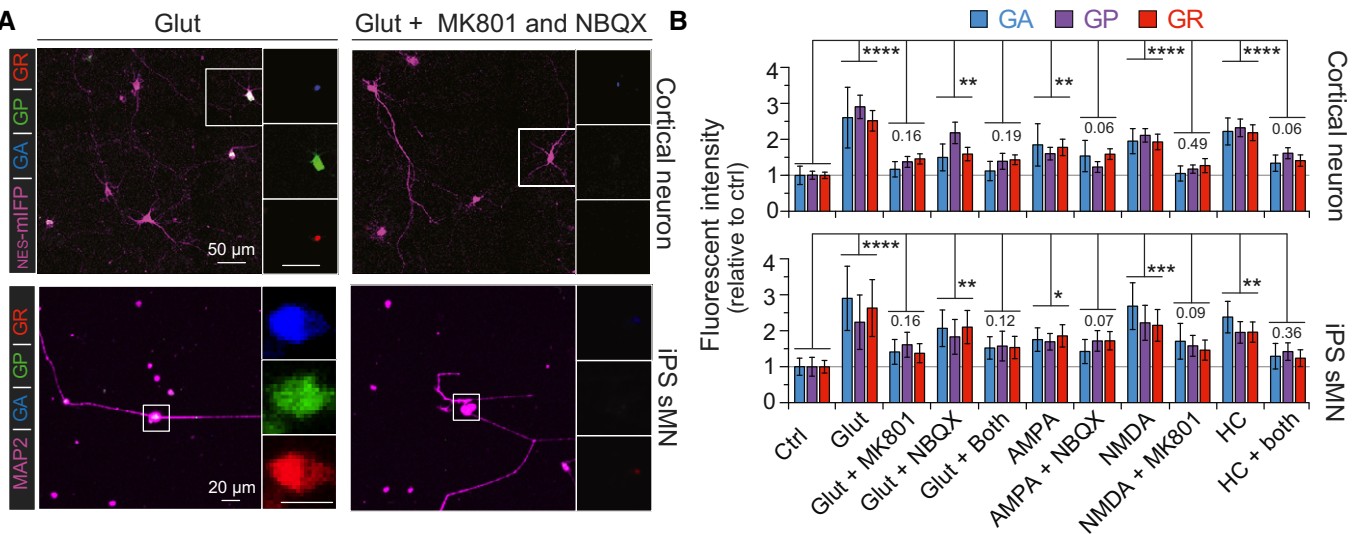

**Figure 3. Excitotoxic stress-induced *C9orf72* NRE-linked DPR levels can be decreased by blocking AMPA or NMDA in primary cortical neurons and in patient-derived iPS spinal motor neurons.**

A Fluorescent microscopy images show glutamate-induced (Glut) excitotoxic stress increases DPR levels in rat cortical neurons expressing the C9 DPR reporter and in iPS sMN derived from *C9orf72* NRE patients. Glutamate-induced DPR levels are reduced by pretreating cells with the AMPA- and NMDA-specific antagonists NBQX and MK801, respectively.

B Relative fluorescent intensity quantification demonstrates there is a significant increase in DPR levels when primary cortical neurons and sMNs are challenged with the excitotoxic stressors: glutamate, AMPA, NMDA, and HC. Blocking either the AMPA or NMDA receptors reduces DPR levels in the presence of excitotoxic stressors. NES-mIFP or MAP2 were used to mark transfection-positive rat cortical neurons or iPS sMNs, respectively, for quantification. DPR fluorescent intensity in iPS sMN was measured using DPR-specific antibodies. Changes in DPR levels are relative to non-stressed cells (CTRL) Cortical neuron: $n = 6$ with $m > 40$ cells per $n$. IPSC sMN: $n = 4$ with $m > 25$ cells per $n$. (****$P < 0.0001$, ***$P < 0.001$, **$P < 0.01$, *$P < 0.05$). All error bars represent the SEM, and all statistics were performed using an uncorrected Fisher's exact test.

Source data are available online for this figure.

translation, reducing DPR levels by ~55%. Therefore, given the more significant response of NMDA activation and inhibition our data indicate excess neuronal calcium signaling may play a role in the production of DPRs through non-AUG-dependent translation (Fig 3A and B). These results demonstrate that glutamate and other known excitotoxic stressors like HC, NMDA, and AMPA can increase DPRs via non-AUG-dependent translation, but AMPA and NMDA channel antagonists can drastically reduce this excitotoxic stress-linked translational phenomenon.

## Repetitive neuronal depolarization increases *C9orf72* NRE non-AUG-dependent translation

Sustained depolarizations in excitable neurons can be triggered by aberrant extracellular accumulation of glutamate, which may lead to excitotoxic mechanisms of neuronal dysfunction and degeneration (Starr & Sattler, 2018). To test whether non-AUG-dependent translation of *C9orf27* NRE is linked and could be modified by neuronal activity, we measured DPR levels in rat CN^NES-mIFP in which we expressed channelrhodopsin-2 to optogenetically control their depolarization–repolarization activity. Channelrhodopsin-2 (ChR2) allows for controlled neuronal depolarization through precise light (470 nm) pulses that allow non-selective permeation of cations through the ChR2, which ultimately leads to neuronal depolarization (Lin, 2011). Using primary rat CN cotransfected with the NES-mIFP, *C9orf72* NRE reporter, and ChR2-YFP, we applied light pulses (at 488 nm) of high- (100 Hz), medium- (10 Hz), and

low-frequency (1 Hz) stimulations (30 s) on neurons expressing ChR2-YFP followed by brief recovery time intervals (25 s) over 20 m, to assess different neuronal depolarization modes (Fig 4A). Since RAN translation in all three ORFs of the C9 reporter construct showed similar relative changes in levels in response to treatments, in the subsequent experiments we utilized only the GP frame, which provides diffuse, uniform fluorescent signal, to assess modulators of RAN translation. Quantification of DPR fluorescence intensity shows a significant increase (~1.6-fold) in DPR production following medium- and high-stimulated neurons compared to low- or non-stimulated neurons (Fig 4B and C). Together, these results indicate that repetitive neuronal depolarizations, which are possibly associated with increased neuronal activity, promote non-AUG translation and therefore lead to a neuron-specific and activity-dependent increase in DPR levels in disease-relevant cells.

## Increased *C9orf72* NRE non-AUG-dependent translation is concomitant with integrated stress response activation

We next examined cellular pathways that may lead to non-AUG-dependent translation of the *C9orf72* NRE. A primary candidate was the integrated stress response (ISR) pathway, since many of the non-AUG-dependent inducing stressors we examined in this study ultimately converge and activate this pathway. The ISR restores cellular homeostasis by activating specific kinases, such as PERK, to phosphorylate eif2α in response to diverse stress stimuli (Pakos Zebrucka *et al*, 2016). For example, phosphorylated eif2α activates

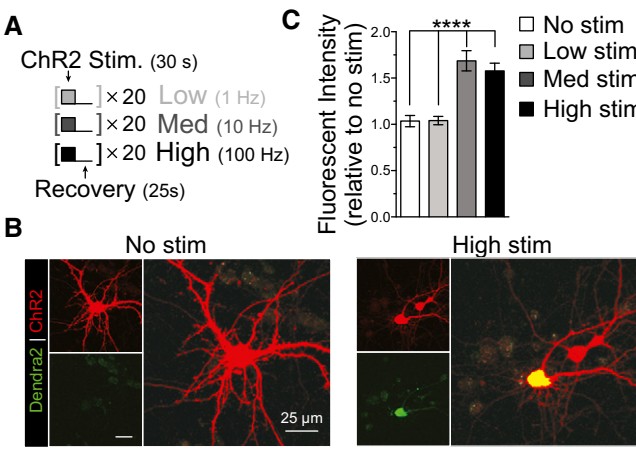

**Figure 4. Increased neuronal activity increases non-AUG-dependent translation of DPRs in primary cortical neurons.**

A  Neuronal activity was induced with Chr2 stimulation by performing trains of LED light pulses at various frequencies: No stim = 0 Hz; Low = 1 Hz; Med = 10 Hz; High = 100 Hz. Trains included 30 s of light pulses followed by 25 s of recovery for desensitized ChR2; trains were performed 20 times.

B  Microscopy images show increased DPR fluorescent intensity levels from medium- and high-frequency stimulation of ChR2 in primary cortical neurons transfected with the C9 DPR reporter.

C  Relative fluorescent intensity quantification shows a significant increase in DPR levels with increases in neuronal activity through medium and high stimulation in primary cortical neurons. No stimulation and low stimulation frequencies did not increase levels of DPR. The C9 DPR reporter, monitoring the GP DPR (ORF2), was cotransfected with ChR2. Cortical neuron: $n = 4$ with $m > 10$ cells per $n$. (****$P < 0.0001$) All error bars represent the SEM, and all statistics were performed using an uncorrected Fisher's exact test.

the transcription factor ATF4 and reduces general translation by preventing the regeneration of the eif2α ternary complex (composed of eif2α, GTP, and an initiation tRNA (Kwon *et al*, 2017). Additionally, neuronal activity-stimulated translation and non-conventional translation of mRNAs are highly dependent on the phosphorylation status of eif2α (Chesnokova *et al*, 2017). Therefore, we measured the levels of critical proteins and modified proteins in the ISR by filter-trap binding and Western blot assays performed on lysates from NSC34 cells transfected with the *C9orf72* DPR reporter to determine whether non-AUG-dependent *C9or7f2* NRE-linked DPR production occurs with ISR activation following stressor treatment. Indeed, the results show significant increases in the ISR-related protein levels for: PERK, a kinase that phosphorylates eif2α; eif2α-P; and ATF4 following stressor treatments (Fig 5A; individual quantifications in Fig EV5B and C) demonstrating ISR activation and DPR levels increase together. We also noted basal cell type-specific differences in ISR-related protein levels, with HEK293T cells showing greater ISR-related protein levels than NSC34 or primary cortical neurons in the absence of additional stressor paradigms. This cell-dependent variability in the ISR could explain the cell type differences in non-AUG-dependent translation levels (Figs 1D, and EV1D and EV5A).

We then further explored whether ISR-related protein levels also increase following the excitotoxic stress paradigm we previously demonstrated increased RAN translation. Rat cortical neurons were cotransfected with *C9orf72* DPR and NES-mIFP reporters, challenged with excitotoxic stressors, and then, ISR-related protein levels were measured using quantitative immunofluorescence imaging. As with

other cellular stressors, PERK and eif2α-P levels increase following excitotoxic stress (Fig 5A). However, this response could be pharmacologically lessened to near control levels using the AMPA and NMDA blockers, NBQX and MK801, respectively. These results implicate that the ionotropic glutamate receptors are involved in activating the ISR and can drive DPR production (Fig 5B).

## Non-AUG-dependent translation of the C9orf72 NRE can be attenuated by therapeutically targeting components of the ISR

We next pursued small-molecule inhibitors that could potentially reduce non-AUG-dependent translation and thereby the production of potentially toxic *C9orf72* NRE-linked DPRs. We identified that various cellular stress events converge on activation of the ISR and result in an increase in non-AUG-dependent translation of DPRs. Moreover, recent work demonstrated that altered levels or modification to ISR-related proteins, such as phosphorylation of PERK or eif2α, leads to altered DPR levels (Green *et al*, 2017; Cheng *et al*, 2018). Therefore, we focused on examining small-molecule inhibitors that target the ISR or translation-related pathways that are currently in use or have been used in clinical trials for neurodegenerative diseases or cancer (Bhat *et al*, 2015) Specifically, trazodone and dibenzoylmethane (1,3 DBM), which recover the eif2α ternary complex and prevent increased ATF4 activation in the ISR, have been shown to be non-toxic compounds that can readily cross the blood–brain barrier, unlike other p-eif2α targeting molecules such as ISRIB. Additionally, in mice, trazodone and 1,3 DBM prevented the disease-linked pathology and rescued memory deficits in prion-diseased and tauopathy FTD mice (Halliday *et al*, 2017). Therefore, to test the ability of these and other translation-targeting candidate compounds in modulating DPR levels, we first pre-incubated NSC34[NES-mIFP] cells with our *C9orf72* DPR reporter with each of the compounds for 1 h prior to the addition of cell stressor, NaArs or TM, followed by quantifying DPR levels 24 h post-stressor treatment. Of the compounds we tested, we identified that a PERK inhibitor (GSK2606414), an eIF4E phosphorylation inhibitor (cercosporamide), trazodone, and 1,3 DBM cause a significant reduction in DPR levels relative to control when challenged with the stressors NaArs or TM (Fig 6A). Reduction of DPR levels through these compounds was independent of changes in transcript levels of the reporter construct (Fig EV1G). These results confirmed previous findings that showed the use of PERK inhibitors may reduce DPR levels in *C9orf72* NRE models (Cheng *et al*, 2018). We then expanded on these findings to test whether these compounds could modulate DPR levels increased by excitotoxicity in primary cortical neurons. We pre-incubated CN[NES-mIFP] cells with our *C9orf72* DPR reporter with each of the compounds for 1 h prior to the addition of glutamate and quantified DPR levels 24 h post-stressor treatment. In cortical neurons, only trazodone and 1,3 DBM caused a significant reduction in DPR levels relative to control (Fig 6B). Importantly, several of the molecules also tested here target the ISR and are currently being examined for cancer or other neurodegenerative diseases. Taken together, our results show that increased non-AUG-dependent translation is mechanistically linked to the ISR and can be decreased by blocking components of the ISR, thus providing a promising therapeutic strategy to mitigate *C9orf72* DPR-linked ALS/FTD pathogenesis.

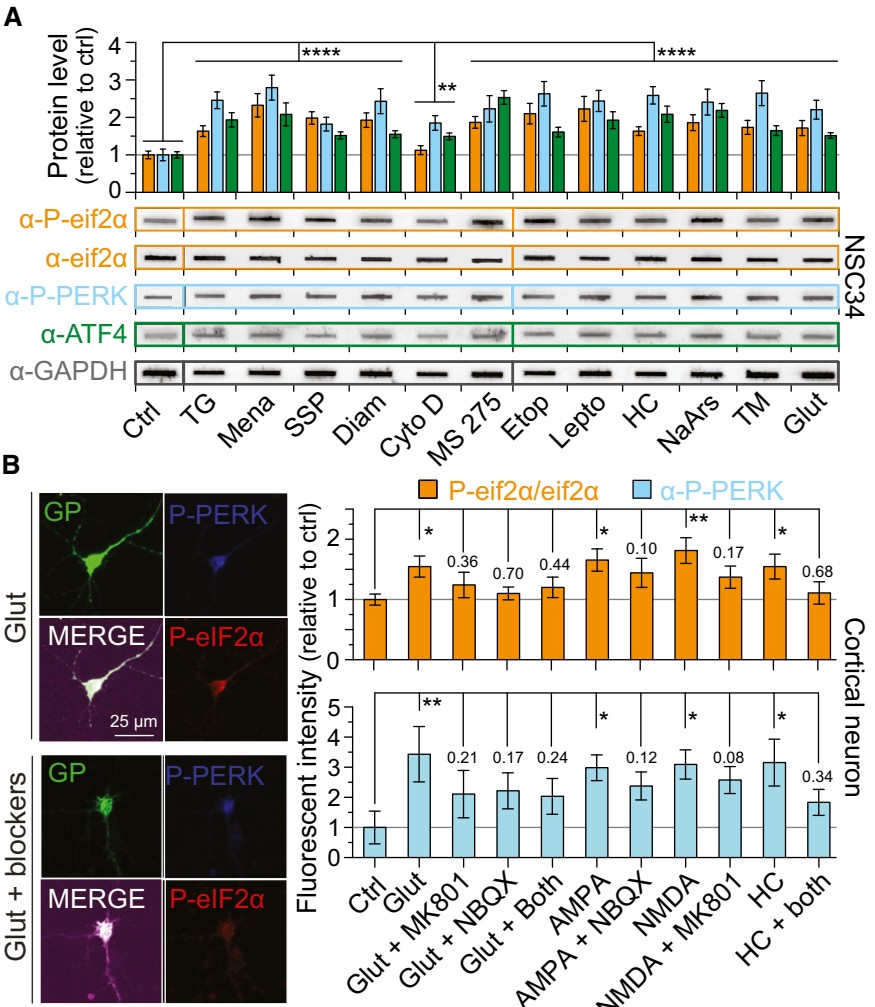

**Figure 5.  Increases in ISR proteins are concomitant with DPR-inducing cellular or excitotoxic stressors.**

A  Filter-trap binding assays show that ISR proteins are enhanced by all stressors. Significant increases following cellular stress treatments are seen in the ratio of phospho-eif2α to total eif2α as well as phospho-PERK and ATF4 levels compared to untreated controls. Changes in ISR proteins are normalized to GAPDH and shown relative to non-stressed or DMSO-only-treated cells (CTRL) depending on the vehicle compounds were dissolved in. $n = 4$.

B  Fluorescent intensity measurements show the levels of two ISR-related proteins, PERK and relative phospho-eif2α (phospho-eif2α/total eif2α), are increased following different excitotoxic stress treatments in cortical neurons. Representative images indicate fluorescent intensity differences for phospho-eif2α and phospho-PERK levels 24 h post-treatment with different excitotoxic stressor paradigms. Blocking of AMPA and NMDA receptors using known antagonists significantly reduces ISR-related protein levels, which parallels reduced DPR levels (Fig 3B). Changes in ISR proteins are relative to non-stressed cells (CTRL). $n = 4$ with $m > 40$ cells per $n$. All error bars represent the SEM, and all statistics were performed using an uncorrected Fisher's exact test.

Data information: ****$P < 0.0001$, **$P < 0.01$, *$P < 0.05$.
Source data are available online for this figure.

## Discussion

NREs have been associated with over 30 neurological and neuro-muscular disorders (Pearson *et al*, 2005; McMurray, 2010). The unconventional non-AUG-dependent translation of the NREs that leads to the production of polypeptides has been identified in many NREs as a pathogenic mechanism (Cleary & Ranum, 2017). In *C9orf72 NRE*-linked disease, the bidirectional transcription of the NRE can lead to the translation of six DPRs that have been shown to be toxic in numerous model systems through various proteinaceous mechanistic origins (Freibaum & Taylor, 2017). To understand

mechanisms that facilitate non-AUG-dependent translation and to potentially reveal new neurodegenerative disease therapeutic opportunities, we developed cellular model systems that allowed us to monitor DPR protein dynamics and levels using fluorescent imaging. Using this *C9orf72* NRE model system, we identified that DPRs are long-lived and DPR levels increase due to non-AUG-dependent translation under conditions of cellular stress. Additionally, cell type-specific differences became more apparent upon our examination of excitotoxic stressors or repetitive neuronal depolarization, which lead to increased DPR levels in neurons. Importantly, the non-AUG -dependent translation of DPRs can be lowered using

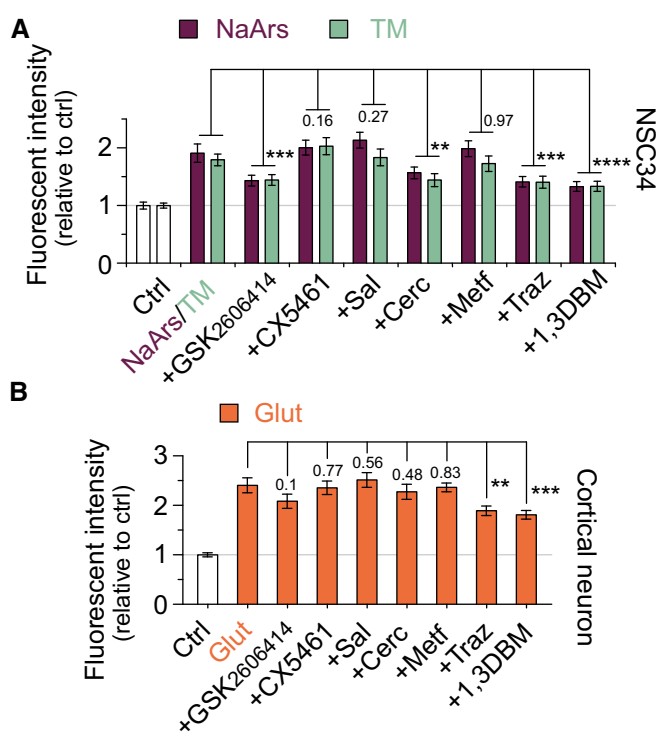

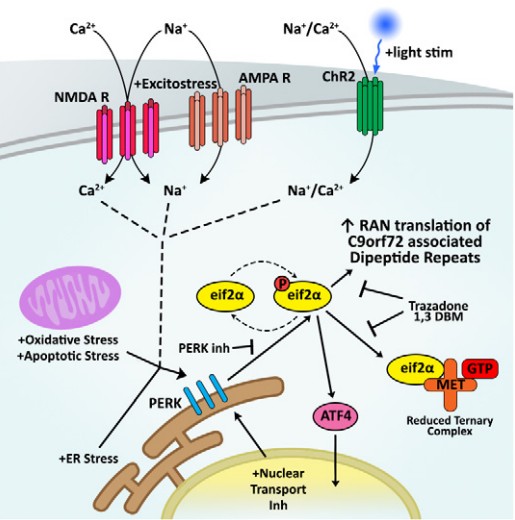

**Figure 7.  Non-AUG translation of DPRs and altered eif2α-P levels occur as a result of cellular stress and increased neuronal activity.**

Numerous cellular stresses and/or increased neuronal activity mechanistically converges to increase phosphorylation of eif2α, and thereby drives the non-AUG-dependent translation of DPRs from *C9orf72* NRE containing transcripts. Phosphorylation of eif2α is a key component of the ISR and is regulated by various ISR-related kinases. One of these kinases, PERK, also shows increased levels in response to cellular stressors, neuronal excitotoxic stress, or aberrant neuronal excitation. Perturbing or blocking specific components of the ISR, using compounds such as the FDA-approved trazodone or dibenzoylmethane (1,3DBM), reduces stress-induced *C9orf72* NRE-linked DPR levels and provides promising therapeutic strategies to combat NRE-linked neurodegeneration.

**Figure 6.   DPR production is inhibited through pharmacological intervention of ISR-related pathways.**

A   Relative fluorescent intensity quantification shows a significant decrease in C9 DPR reporter levels when stressor-challenged NSC34 cells were treated with compounds inhibiting ISR-related or translation-targeting pathways. Dibenzoylmethane (1,3DBM; 20 μM), which inhibits eif2α-P activity, had the most significant decreases in DPR production. Trazodone (Traz; 20 μM) showed similar significant reduction in DPR levels as the PERK inhibitor, GSK2606414 (500 nM).

B   Relative fluorescent intensity quantification shows a significant decrease in C9 DPR reporter levels when glutamate-stressed primary cortical neurons were treated with the compounds trazodone (Traz; 20 μM) or dibenzoylmethane (1,3DBM; 20 μM). Similar to the treatments in NSC34, 1,3DBM shows the most significant decrease in DPR production.

Data information: Changes in DPR levels are relative to DMSO-only-treated cells (CTRL). GSK2606414 (500 nM), CX5461 (250 nM), salubrinal (Sal; 15 μM), cercosporamide (Cerco; 120 nM), metformin (Metf; 500 μM), trazodone (Traz; 20 μM), dibenzoylmethane (1,3DBM; 20 μM). NSC34: $n = 4$ with $m > 100$ cells per $n$. Cortical neuron: $n = 4$ with $m > 40$ cells per $n$. ****$P < 0.0001$, ***$P < 0.001$, **$P < 0.01$.  All error bars represent the SEM, and all statistics were performed using an uncorrected Fisher's exact test.

ionotropic glutamate receptor antagonists or using small-molecule inhibitors targeting the ISR—a convergent pathway we identified through our systematic DPR-inducing analyses (Fig 7). Therefore, this work provides early findings for new therapeutic avenues to ameliorate toxic polypeptide production caused by the non-AUG-dependent translation of NREs, which is a proposed pathogenic mechanism identified for a number of neurological or neuromuscular disorders.

Our findings extend recent independent non-AUG-dependent mechanistic studies that have demonstrated a relationship between activation of the ISR and increased non-AUG-dependent translation of the *C9orf72* NRE (Green *et al*, 2017; Cheng *et al*, 2018; Sonobe *et al*, 2018). The ISR decreases canonical translation while levels of p-eif2α,

PERK, and ATF4 are increased. Utilizing various stressors related to aging, such as oxidative stress, ER stress, or DNA damage, we demonstrate a significant concomitant increase in non-AUG translation of DPRs and activation of the ISR. Consistent with recent findings (Green *et al*, 2017; Cheng *et al*, 2018; Sonobe *et al*, 2018), our results indicate that stress-induced dysregulation of the ISR can lead to an increase in *C9orf72* NRE DPR production, which we show have slow turnover rates. Additionally, dysregulation of ISR-related proteins, such as p-eif2α, has been identified in many age-related neurodegenerative disorders (Kourtis & Tavernarakis, 2011). Together, this suggests that the combination of increased age-dependent stimulation of the ISR and slow DPR turnover rates could exacerbate a positive feedback loop for non-AUG-dependent DPR translation, which may lead to rapidly accumulating DPR levels that accelerate disease (Cheng *et al*, 2018). Furthermore, cell-specific sensitivity or tolerance to stress may determine the rates at which these DPRs levels accumulate in parallel with ISR activation. For example, we show that the ISR-associated proteins are inherently upregulated in HEK293T cells versus NSC34 or cortical neurons, and HEK293T cells robustly express DPRs. Therefore, based on our findings, cell types and possibly cell age are important considerations when modeling mechanisms and/or drivers of non-AUG-dependent translation for diseases that are primarily neurological and neuromuscular disorders.

There is increasing evidence that under conditions of neuronal stress or aberrant neuronal activity there is a translational shift to a subset of RNAs. These differentially translated RNA subsets often contain features also shared with *C9orf72* NRE mutation containing RNAs: (i) a guanine- and cytosine-rich sequences present in the 5′

UTR that can form nucleic acid secondary structures (Fratta *et al*, 2012; Reddy *et al*, 2013; Haeusler *et al*, 2014); and (ii) an upstream AUG-initiated ORFs (Tabet *et al*, 2018), which serves as regulatory elements that alter downstream translation of the AUG-initiated ORFs in response to neuronal stress or hyperactivity (Chesnokova *et al*, 2017). Therefore, given the *C9orf72* gene transcripts are most abundant in the brain of human post-mortem tissue (Carithers *et al*, 2015), it may not be surprising that *C9orf72* NRE DPRs are more readily detected within these regions that are directly linked to neurodegenerative disease (Saberi *et al*, 2018). Furthermore, increased glutamate receptor levels and age-dependent aberrant neuronal excitability have been observed in *C9orf72* NRE-linked ALS/FTD *in vitro* models, animal models, and/or patients (Selvaraj *et al*, 2018; Starr:2018fv Shi *et al*, 2018), and we demonstrate through our neuronal excitotoxic stress and neuronal activity experiments that increased neuronal excitability increases non-AUG-dependent translation of the *C9orf72* NRE. Therefore, these results suggest that the combination of abundant *C9orf72* NRE containing transcripts, sustained depolarization/repolarization cycles" which potentially are associated to excitotoxic stress, and ISR activation may lead to the production of proteinaceous DPRs that drive ALS/FTD disease progression primarily in neurons.

The investigation into neuronal drivers of the non-AUG-dependent translation of the *C9orf72* NRE provides new mechanistic opportunities to modulate DPR production. Using AMPA or NMDA receptor antagonists MK801 or NBQX, respectively, we demonstrated that DPR levels could be reduced in neurons under conditions of excitotoxic stress. These results suggest that cellular influx of calcium may be a signal that stimulates the ISR and together triggers non-AUG-dependent translation, which has important implications in age-dependent NRE-linked diseases. However, further studies are required to determine the mechanistic contribution of ionotropic and/or metabotropic receptors, and the signaling cascades that are involved in modulating neuronal activity-associated polypeptide production and in NRE-linked neurodegeneration.

Finally, our therapeutic investigation into pharmacological inhibitors of non-AUG-dependent translation that target the ISR pathways provided novel non-AUG-dependent mechanistic insight and meaningful therapeutic strategies to reduce DPR translation. In this study, we identified two highly promising FDA-approved drugs, trazodone and dibenzoylmethane, that are able to reduce DPR levels. These two drugs readily cross the blood–brain barrier, have little toxicity, have been previously utilized to treat depression, and are being repurposed for treating diseases such as Alzheimer disease (Halliday *et al*, 2017). In summary, this work identified drivers of and cell type-specific differences for non-AUG-dependent translation of the ALS/FTD-linked *C9orf72* NRE, and we uncovered novel therapeutic candidates that may be quickly repurposed to begin treating neurodegeneration caused by non-AUG-dependent translation of NRE mutations.

# Materials and Methods

## Plasmid construct design

All constructs used in this work were assembled in pcDNA3.0/pcDNA3.1 (Invitrogen/Thermo Fisher) or pAAVS1 (Addgene #66577) plasmid backbones. The constructs are included as dna files (see Dataset EV1), which contain the assembly strategy history, the primers (IDT) and enzymes (NEB) used to generate inserts via classical restriction digest cloning or Gibson assembly cloning (using the NEBuilder online tool) following manufacturers' recommendations. The 5′ gene region for *C9orf72* was PCR amplified from purified HEK293 genomic DNA. Dendra2 construct was cloned from and plasmid obtained from Addgene (#51005). The mIFP construct was obtained through a generous gift from Dr. Christopher Donnelly and cloned into the pCDNA3.0 backbone. All PCR amplification was performed using standard Phusion/Q5 polymerase protocols (NEB), and the amplified products were purified using affinity columns or through agarose gel extraction following manufacturer's protocols (Fisher). All repeats were inserted between BamHI and XhoI sites in the plasmid backbones using restriction digest cloning. The *C9orf72* NRE repeats, $(G_4C_2)_{\sim 188}$, were generated following previously described protocols (Wen *et al*, 2014). The bacterial origin of replication for the plasmid was reoriented relative to the direction for the antisense repeats to reduce instability of the *C9orf72* NRE inserts (Trinh & Sinden, 1991).

## Cell cultures and transfections

For assessment of RAN translation of the $(G_4C_2)_{188}$-Dendra2-HA or $(C_4G_2)_{188}$-Dendra2-HA constructs, HEK293T cells (obtained from ATCC) were seeded into 24-well glass bottom plates (Cellvis P24-1.5H-N) at a density of $1 \times 10^5$ cells/well and transfected using Lipofectamine 2000 (Invitrogen) with 1 μg of total DNA/well following manufacturer's recommendation. HEK293T cells were maintained in DMEM supplemented with 10% FBS (Life Technologies; 37°C, 5% $CO_2$) and 1× penicillin/streptomycin. In all cases, a 4:1 DNA ratio was used for the reporter constructs to NES-mIFP based on DPR: Reporter protein expression optimization previously determined (Wen *et al*, 2014).

For assessment of stress-induced RAN translation, NSC34, primary rat cortical neurons, and sMNs derived from human iPSC were utilized. NSC34 were seeded into 24-well glass bottom plates at a density of $1 \times 10^5$ cells/well and transfected using Lipofectamine 2000 (1 μg of total DNA/well; 4:1 DNA ratio). NSC34 were maintained in DMEM containing 10% FBS (Life Technologies; 37°C, 5% $CO_2$). Primary cortical neurons, prepared from E19 Sprague Dawley rat embryos as previously described (Kayser *et al*, 2006), were plated ($1 \times 10^5$ cells/well) on poly-D-lysine and laminin-coated 24-well plates and transfected using Lipofectamine 2000 (Invitrogen, DIV 10, 1 μg of total DNA/well; 4:1 DNA ratio). Human iPSCs were propagated in the Jefferson Stem Cell Center and then provided to us for differentiation into sMNs using a previously described protocol (Maury *et al*, 2015). The *C9orf72* iPSC lines were acquired through Target ALS (ID: TALS9-9.3 and TALS 9-9.5).

For fluorescent quantification of DPR levels between cell types, fluorescent levels of the C9 DPR reporter construct were normalized to fluorescent levels of NES-mIFP. This approach was used to adjust for the inherent cell-specific differences in AUG-dependent translation.

## Stress-induced DPR comparisons

The following compounds were utilized to induce various stresses: thapsigargin (#T9033), tunicamycin (#T7765), allyl alcohol (#240532),

menadione (#5750), bromobenzene (#16350), staurosporine (#S5921), cytochalasin D (#C8273), diamide (Santa Cruz #sc-211289), MS-275 (Santa Cruz #sc-279455A), etoposide (#E1383), leptomycin B (Cell Signaling #9676S), homocystine (#69453), paraquat (#36541), glutamate (#G1626), $H_2O_2$ (#HX0640), and sodium arsenite (#S7400) (all compounds from Sigma unless specified). A dose–response curve was performed to determine $EC_{50}$ values for production of DPRs for each stressor. For NSC34 and non-excitotoxic stressors in neurons, cells were incubated with stressors for 24 h before assays were performed. For excitotoxic stress studies in neurons, cells were incubated in the presence of stressors for 2 h, after which the cells were washed with respective culture media and analyzed 24 h later.

The following compounds (from Sigma unless specified) were utilized for inhibition of RAN translation: MK801 (#M107; 1 μM), NBQX (#N171; 1 μM), GSK2606414 (#516535; 500 nM), CX5461 (Millipore #509265; 250 nM), salubrinal (#SML0951; 15 μM), cercosporamide (Tocris #4500; 120 nM), metformin hydrochloride (Tocris #2864; 500 μM), trazodone (#T6154; 20 μM), and dibenzoylmethane (#D33454; 20 μM). Cells were pre-incubated with inhibitors for 1 h and then incubated with both the inhibitor and stressors for 24 h.

For fluorescent quantification of DPR levels in HEK, NSC34, and primary cortical neurons, only cells that were positive for the NES-mIFP construct were measured to adjust for transfection efficiency and exclude cells that did not contain the C9 DPR reporter construct. Quantification of changes in DPR levels in all cell types is shown relative to non-stressed or DMSO-only-treated cells (CTRL) depending on the vehicle compounds were dissolved in. For comparison of various conditions within the same cell type, fluorescent intensity is not normalized to fluorescent levels of NES-mIFP.

### Induction of neuronal activity through channelrhodopsin2

Dissociated cortical neurons were transfected with the following plasmids: NRE-Dendra, hChR2(H134R)-mCherry (Addgene #20938), our $(G_4C_2)_{188}$-GP-Dendra2-HA reporter construct, and our NES-mIFP construct (1:4:1 DNA ratio, respectively). 48 h post-transfection, trains of activity were induced in cortical neurons through ChR2 activation using 488 nm blue LED light with an estimated light intensity of ~15 mW/cm$^2$ (Amuza LEDA4-B). Four stimulation protocols were used: 100, 10, 1 Hz, and no stimulation. Light flashes were delivered at these frequencies, respectively, in 30-s trains with 25-s re-sensitization intervals. A total of 20 trains were delivered consecutively. To confirm neuronal activity and calcium chelation after blue light stimulation, cortical neurons were transfected with the fluorescent calcium indicator rCaMP1 h (Addgene #42874) along with ChR2 constructs. rCaMP1 h fluorescence was imaged under fast time-lapse imaging (1 frame/s) simultaneous to ChR2 light stimulation using confocal imaging (Nikon A1R), and calcium fluctuations were assessed.

### Western blots and filter-trap binding assays

The detection of DPRs and internal stress response-related proteins was performed in cells lysed in RIPA buffer and then blotted to a nitrocellulose membrane with 0.2 μm pore size (Bio-Rad) in a slot blot apparatus. Dot blot detection of DPRs utilized the same procedure without the use of the slot blot apparatus. Quantification of band intensity for Western blots and filter-trap binding assays

was performed using ImageJ. Protein levels were normalized to GAPDH. The following primary antibodies were used at indicated dilutions: anti-HA (Abcam Cat# ab18181, RRID:AB_444303; 1/1,000), anti-GA (MABN889; 0.05 μg/ml), anti-GP (ABN1358; 1/1,000), anti-GR (MABN778; 0.2 μg/ml), anti-pPERK (ab192591; 1/1,000), anti-peif2α (Abcam Cat# ab32157, RRID:AB_732117; 1/500), anti-eif2α (Santa Cruz Biotechnology Cat# sc-133132, RRID: AB_1562699; 1/1,000), anti-ATF4 (Cell Signaling Technology Cat# 11815S, RRID:AB_2616025; 1/1,000), anti-ATF6 (Abcam Cat# ab122897, RRID:AB_10899171; 1/500), and anti-GAPDH (Fitzgerald Industries International Cat# 10R-G109a, RRID:AB_1285808; 1/10,000).

### Immunofluorescence and confocal microscopy

Cells were fixed in 4% PFA in PBS (15 m at room temperature), permeabilized in 0.3% PBS-T (15 min), blocked in 1% BSA in PBS-T (1 h at room temperature), incubated with primary antibodies (overnight at 4°C), secondary antibodies (1 h at room temperature), and mounted in DAPI-free anti-fade mounting media. Cells were washed three times with PBS between each step and imaged by confocal microscopy (Nikon A1R; utilizing 10× and 20× objectives) at a 0.3 μm step size. An average of 16 fields (2–5 neurons and 30 non-neuronal) were imaged per well. Representative images are shown from at least three separate experiments. Automated thresholding through NES elements and ImageJ was used to correct for background fluorescence and expedite quantification. The values used remained constant for individual experimental *n* to keep quantification consistent through each experimental condition. MAP2-positive neuronal cells were identified using 1:500 dilution of anti-MAP2 (Millipore Cat# AB5622, RRID:AB_91939) following the immunofluorescence (IF) procedures described above. Staining for anti-GA, anti-GP, and anti-GR utilized the same antibodies and concentrations as mentioned above.

### Determining of DPR half-life

For the longitudinal DPR half-life analyses, HEK293T cells were seeded on polyethyleneimine (PEI)-coated 24-well glass bottom plates at a density of $2.5 \times 10^4$ cells/well. The combination of increased plate adherence and sparser cell plating density were optimal for single-cell tracking and quantification. HEK293T cells were transfected with the $(G_4C_2)_{188}$-Dendra2-HA or $(C_4G_2)_{188}$-Dendra2-HA reporter constructs 24 h post-plating. Twenty-four hours post-transfection, cells were placed in an incubator chamber inserted under the Nikon A1R to allow cell viability over long imaging experiments. Wells were then visualized to find transfection-positive HEK293T cells that were isolated for expedited monitoring and quantification over time. Photoconversion from Dendra2-green to Dendra2-red was performed using successive 405 nm irradiations for 15 s. Photoconverted DPRs were followed for a total of 108 h and quantified using NIS-elements.

### Quantitative real-time PCR analysis

Measurements of DPR-Dendra2 and NES-mIFP mRNA levels were made by qRT–PCR. For cell-specific measurements of transcript levels, total RNA was extracted 24 h post-transfection using the

## The paper explained

### Problem

Nucleotide repeat expansions (NREs) have been associated with over 30 neurological and neuromuscular disorders with the non-AUG-dependent translation of the NRE emerging as a potential common pathogenic feature of many of these diseases. The pathological features that result from non-AUG- dependent translation have been well-catalogued in disease-relevant tissue from patients carrying an NRE mutation. However, the cellular and cell type-specific mechanisms that increase the usage of non-AUG- dependent translation are still not well understood. Our work examines these mechanisms in disease-relevant cells, neurons. This leads to the accumulation of toxic polypeptides in cells, and this accumulation increases in response to stressful cellular stimuli.

### Results

Using the *C9orf72* NRE-linked neurodegenerative paradigm, we demonstrate in neurons that excitotoxic stress and increased neuronal excitation drive the production of polypeptides through RAN translation. This result alone has important pathogenic implications in cell models relevant to neurodegenerative disease, since *C9orf72* NRE patient-derived iPS motor neurons show age-dependent hyperexcitability or hypoexcitability. Additionally, through our rigorous investigation, we uncovered new mechanistic insights into the relationship between RAN translation and activation of the integrated stress response (ISR). Importantly, we also demonstrate that we can modify RAN translation using FDA-approved compounds that target components of the ISR and have been previously administered in clinical trials to treat other neurological and neuromuscular disorders.

### Impact

Our results provide new mechanistic understanding for the role of RAN translation in driving disease pathogenesis and provide new therapeutic approaches to attenuate NRE-linked neurodegeneration that can be tested in the clinic immediately.

PureLink™ RNA Mini Kit (Invitrogen). For measurement of transcript levels in stressed cells, cells were incubated with stressors 24 h before total RNA was extracted. The cDNA was generated from total RNA through a standard reverse transcription protocol using the reverse transcriptase SuperScript IV (Invitrogen). Primer sets were designed and optimized to recognize Dendra2 or mIFP. The optimal primer sets used were as follows: Dendra2—forward (TCAGCTACGACATCCTGACC) and reverse (CGTTCTGGAAGAAG CAGTCG); NES-mIFP—forward (GCGTCGTGTGGACTGTACC) and reverse (CGATCTGCGGAATAGTAGACG). The qRT–PCR was performed using a PowerSYBR® Green PCR Master Mix (Thermo) and following manufacturer's recommended thermocycler settings. Fluorescence was measured, and $C_T$ values were analyzed using the QuantStudio 5 Real-Time PCR system (ThermoScientific). To account for potential transfection efficiency differences, Dendra2 $C_T$ values were normalized to NES-mIFP $C_T$ values.

### Image analyses and statistical analyses

Image analysis was performed using NIS-elements and ImageJ. In these programs, threshold adjustments, separation of cell areas, and fluorescent quantification were automated and kept constant through individual experimental groups to maintain consistent parameters for imaging analysis. Experiments utilized a four or

greater sample size to ensure that experiments were reproducible and statistical differences were accurate. Some supplementary experiments (such as longitudinal analysis of half-life) utilized a three-sample size due to the length of the experimental procedures. However, this still allowed for statistical differences to be observed. All data were included during quantification. Outliers were then calculated using a > 3 standard deviations away from mean and excluded from statistical comparisons. One-way and two-way ANOVAs (depending on experimental setup) were utilized to compare across multiple treatments. Specifically, an uncorrected Fisher LSD two-way ANOVA was primarily used to compare the three DPRs for CTRL conditions versus treatment conditions unless otherwise noted in the figure legend. All statistical analyses were performed in GraphPad Prism, and all statistical parameters, outputs, and *P* values are included in Appendix Table S1.

**Expanded View** for this article is available online.

## Acknowledgements

We thank the Stem Cell Center of the Vickie and Jack Farber Institute for Neuroscience for propagating and supplying patient-derived iPSC. The Stem Cell Center of the Vickie and Jack Farber Institute for Neuroscience is supported through the generosity of Kimberley Strauss and the Strauss Foundation. We thank Dr. Matthew Dalva (Jefferson University) for sharing rat primary cortical neuron preps. NIH (R21-NS090912 to D.T.) (4R00NS091486-03 to A.H.), Muscular Dystrophy Association (to D.T.), the Robert Packard Center for ALS Research (to D.T.), and Farber Family Foundation (to P.P.) grants supported this work.

## Author contributions

Conceptualization, TW, KM, XW, AH, DT; Methodology, TW, KM, KR, XW, YP, AH, DT; Investigation, TW, KM, KR, XW, YP, BM, AH; Formal analysis, TW, KM, AH; Visualization, TW, AH; Writing—original draft, TW, KM, AH, DT, PP; Funding acquisition, AH, DT, PP; Resources, KR, XW, YP, BM, AH; Supervision, AH, DT, PP.

## Conflict of interest

The authors declare that they have no conflict of interest.

## For more information

(i)    Jefferson Weinberg ALS Center: https://hospitals.jefferson.edu/departme nts-and-services/weinberg-als-center.html
(ii)    Aaron Haeusler: https://www.haeuslerlab.com
(iii)    Davide Trotti: https://www.jefferson.edu/university/life-sciences/faculty-staff/faculty/trotti.html
(iv)    C9orf72 gene: https://ghr.nlm.nih.gov/gene/C9orf72#resources.

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
