## [Review Process File · EMBO Molecular Medicine]

Repeat-associated non-AUG translation in C9orf72-ALS/FTD is driven by neuronal excitation and stress

Thomas Westergard, Kevin McAvoy, Katelyn Russell, Xinmei Wen, Yu Pang, Brandie Morris, Piera Pasinelli, Davide Trotti, and Aaron Haeusler

Review timeline:	Submission date:	12 June 2018
	Editorial Decision:	23 July 2018
	Revision received:	24 October 2018
	Editorial Decision:	14 November 2018
	Revision received:	29 November 2018
	Accepted:	6 December 2018

Editor: Céline Carret

Transaction Report:

1st Editorial Decision

23 July 2018

Thank you for the submission of your manuscript to EMBO Molecular Medicine. We have now heard back from the three referees whom we asked to evaluate your manuscript.

You will see from the comments below that while the referees find the manuscript to be of interest, providing novelty and clinical value, they also share similar concerns about artificial errors, quantification criteria, and references. They also suggest experiments to strengthen the results and make them more conclusive. Upon our cross-commenting exercise, referee 3 added, "As this is competitive research field, the authors need to provide careful and convincing data in the manuscript. Otherwise, it will just confuse not only researchers but clinicians and their patients affected by ALS/FTLD."

We would therefore welcome the submission of a revised version within three months for further consideration and would like to encourage you to address all the criticisms raised as suggested to improve conclusiveness and clarity. Please note that EMBO Molecular Medicine strongly supports a single round of revision and that, as acceptance or rejection of the manuscript will depend on another round of review, your responses should be as complete as possible.

I look forward to receiving your revised manuscript.

***** Reviewer's comments *****

Referee #1 (Comments on Novelty/Model System for Author):

Model systems are adequate

Referee #1 (Remarks for Author):

Hexanucleotide repeat expansions of GGGGCC in the C9orf72 gene are the most common genetic cause of ALS and FTD. There are three hypotheses underlying potential disease mechanisms: 1) haploinsufficiency of C9orf72 protein, 2) gain of toxicity from sense and antisense RNA containing these repeats, and 3) non-canonical translation of these repeats leading to toxic dipeptide repeat species. Numerous studies have demonstrated that arginine-rich species (glycine-arginine and proline-arginine) are acutely cytotoxic and are therefore hypothesized to be the major contributors to disease. A major challenge in the field has been to uncover precisely how this non-canonical translation occurs and therapeutic avenues to prevent translation of these toxic dipeptide repeat species.

Westergard et al. recapitulate work from previous groups illustrating that the non-canonical translation of C9orf72 repeats is upregulated by various stressors using a novel dendra2 reporter system. This work supports the hypothesis that the integrated stress response (ISR) can upregulate non-canonical translation of C9orf72 repeats through phosphorylation of eIF2a. Importantly, the authors provide two FDA approved therapeutics trazodone and 1,3 DBM that target eIF2a phosphorylation.

A novel finding facilitated by their photo-convertible dendra2 reporter, was that the dipeptide repeat species are highly stable. In addition, the authors provide evidence that excitotoxic stress can also upregulate translation of C9orf72 repeats. Pharmacologically inhibiting AMPA or NMDA receptors was sufficient to reduce the levels of translation during excitotoxic stress, suggesting an importance for cationic influx as an early step to ISR activation.

Overall the work can be divided into 1) supporting previous work published by Green et al. 2017, Chang et al. 2018 and Sonobe et al. 2018 implicating cellular stress to upregulation of C9orf72 repeat translation and 2) additional work indicating that excitotoxic stress in neuronal cells upregulates C9orf72 repeat translation. This work collectively adds to our understanding of RAN translation mechanisms and is an important contribution. There are additional experiments and controls that should be considered prior to publication and are described in detail below.

Major concerns:

- 1) The authors do a nice job illustrating that despite which frame dendra2 is placed, all three potential dipeptide species can be detected (in order to illustrate that the addition of dendra2 does not influence non-canonical translation of other frames). They also show that dipeptide fusions recapitulate cellular localization of AUG-driven, codon-optimized DPRs. Based on this latter data, they claim that dendra2 is tagging specific DPRs.
 - a. Based off recent work from Tabet et al. (2018), there is evidence that frameshifting can occur with initiation starting at CUG, leading to additional +1 and +2 frame products. Given this data, indicating consistent localization is not sufficient to support the authors' conclusion. The proper experiment would be to do an HA-immunoprecipitation and subsequently probe for each of the three dipeptide species to confirm immunoreactivity with the DPR species being tagged. This would also help to provide more evidence for or against frameshifting.
 - b. Dot blots were only performed for all three DPRs when "ORF3" was tagged. Where is the data for the other two tagged frames? This should be included for completeness of the figure.

- 2) DPR fluorescence quantifications & slot blots. The authors panel an impressive number of drugs in order to assay many different types of cellular stresses. However, in all quantifications of both fluorescence and slot blots there is no clear control that the levels of DPRs are compared against.
 - a. In the Figure S1 panel E, it is indicated that NES-mIFP is used as both a positive transfection marker and a readout of AUG-dependent translation. It is not clear from the method section, within figures, nor in figure legends whether or not the total level of DPRs has been normalized or even

compared to AUG-dependent translation, although it is abundantly clear that NES-mIFP is being utilized as a transfection marker and cytosolic marker.

- i. To resolve this, the authors should include AUG-NES-mIFP (or another AUG-driven protein) expression levels in each experiment. Showing the levels of a control construct side-by-side to DPR levels would be more helpful than the normalization (see note ii below).
- ii. The authors should make it clear in the methods section or figure legends that DPR levels have first been normalized to an AUG-dependent product and then graphed as fold-change relative to "ctrl" treated expression (If this is the case, which I'm assuming, but again not clear).
- iii. It would also be ideal to have data on the mRNA levels of the reporter in each of the stressors to indicate that C9 reporter RNA increase does not account for the increase in DPR levels. At the very least, this must be included for the excitotoxic stress experiments.

3) The authors suggest that calcium influx may be a signal that stimulates the ISR. Their conclusion is that "AMPA and NMDA channel antagonists can drastically reduce...excitotoxic stress-linked [translation]" and that excitotoxic stress can activate ISR as shown by increase in PERK and eIF2a-P levels.

a. A good experiment that is missing here is to repeat the excitotoxic stress and inhibit eIF2a phosphorylation or PERK activation (with drugs used in figure 7) and determine whether DPR levels are increased or reduced. If reduced consistent with the authors' model, this would help to really tie together the idea that excitotoxic stress through AMPA and NMDA receptors activates ISR. So far, I am not convinced the authors have provided sufficient data to conclude that excitotoxic stress converges on the ISR (suggested in text and figure 7).

4) It is interesting that repetitive stimulation of primary neurons leads to an increase in DPR levels. However, figure 4 needs work:

- a. Why is only the GP frame data included here? What about at least the other two sense products, GA and GR?
- b. This figure and authors conclusions "results indicate that repetitive neuronal depolarizations, which are possibly associated with increased neuronal activity promotes non-AUG translation" and further discussions give the overall sense that figure 4 is a weak link currently in the paper. The ALS field (as the authors point out the introduction) is decidedly at odds regarding whether hyper or hypoexcitability is occurring in ALS models. It is an interesting result, however I recommend moving it to the supplement if the experiment is not repeated for GA and GR (as recommend above).

5) Figure 1D, how do the authors know that the differences in # of DPR positive cells and overall fluorescent intensity doesn't simply correspond to transfection efficiency (i.e. HEK293T cells are very well transfected). It's unclear again if only cells with NES-mIFP positive cells are counted, but even so the NES-mIFP reporter must be much smaller than the C9 reporters. For cells that are easy to transfect like HEKs, both reporters probably get taken up without much issue but you might find that the C9 reporters are harder to transfect into primary neurons.

- a. Also, CMV promoter is not the best for expression in neurons, so how can you compare expression in HEKs (where CMV works very well) to neuronal expression?
 - i. Perhaps you can repeat with another promoter such as EF1a that work well in both HEK cells and neuronal cells. This also comes back to point 2iii, controlling for RNA expression.

Minor points:

1. Introduction pg. 2-3: "One potential outcome...dynamic transcriptional regulation...P-eIF2a dependent transcriptional regulation is a key response element to cellular stress."

a. Is the use transcription above a typo? Chesnokova et al. 2017 refer to translational changes caused by eIF2a phosphorylation.

2. Where are c9ALS patient-derived sMN data combined? It's well-established that different c9ALS lines express DPRs are different levels. It would be ideal in the supplement to see the lines separated.

a. Unclear from IF methods whether DPR antibodies used in immunoblots are used again for IF (and if so, dilutions, etc. are missing).

3. Half-life calculations:

a. DPR half-life calculations don't even extend long enough for half the fluorescence to decrease. Is it really appropriate to calculate half-life without at least reaching this point?

4. Overall microscopy images are very small and difficult to see (especially neuronal primary cells or patient cell data).

a. Figure 1 and other cultured cells images are frequently oversaturated. It isn't appropriate to use oversaturated images for quantifications.

5. Figure 2 A, please be clear which DPR you are using.

6. A small typo: Figure 3 A, "...the C9 DPR reporter andi in iPS...."

7. Figure S7 is a nice addition, is it possible to get the one-way tests for DPR levels as well?

a. This could be helpful for better understanding if a specific type of stress or class of stressors can be selective with regards to different ORFs.

8. Should add Sonobe et al. 2018 to references of previously testing RAN translation and cellular stress (in addition to Greene et al. 2017 and Cheng et al. 2018).

9. Figure 5B, is atf4 data missing here or is there an error in including it in the panel legend?

Referee #2 (Remarks for Author):

In the present manuscript, Westergard and colleagues assess the effect of neuronal excitation and cellular stress on RAN translation of 188 G4C2 repeats by expressing Dendra2-tagged DPR constructs in NSC34 cells, primary rat cortical neurons and C9 human iPSC-derived sMNs.

At the time of this review, to our knowledge, three other publications have shown the integrated stress response (ISR) to upregulate G4C2-RAN translation (Green et al., 2017, Cheng et al. 2018, and Sonobe et al., 2018). However, the present manuscript expands upon these previous publications in four key ways. 1) The researchers thoroughly assess the effects of a wider panel of cell stressors, including NMDA receptor mediated excitotoxicity, on C9 RAN translation than previous reports, implicating new stress pathways in this phenomenon. 2) Using C9 patient-derived iPSC sMNs, they show that cell stress increases endogenous DPR levels. 3) They specifically establish a role for excitotoxicity in enhancing C9 RAN translation. 4) They identify two small molecules that reduce the increase in C9 RAN translation under conditions of exogenous cellular stress. Overall, these findings represent an advance from previous reports on how cell stress enhances C9 RAN translation, and of factors that may promote disease in patients. A more significant advance could be potentially achieved with further studies on the newly identified suppressors of stress-induced C9 RAN translation (trazodone and 1,3DBM). has a few shortcomings that need to be addressed.

Concerns (in order of content):

1. In the abstract, the authors state that PERK inhibitors and other compounds "greatly reduce DPR levels." This statement is not supported by the data in the paper. In figure 6, there is a relative reduction in ability of stress to enhance DPR reporter expression, but this is likely not a reduction at all (it is not possible to interpret from the multiply normalized data in this figure) and it is certainly not a "great" reduction.

2. On pages 2-3, in the second to last paragraph of the introduction, the authors should clarify that altered translation is a direct consequence of eIF2 α phosphorylation, with transcriptional changes occurring as a result of altered translation of transcription factors such as ATF4.

3. In Figure 1B-C, the authors use filter trap assays with an anti-HA antibody and Dendra2 fluorescence as readouts for GA, GP, and GR RAN translation from reporter constructs. However, it is possible that Dendra2-HA is expressed independent of RAN translation. To clearly show that Dendra2-HA is fused to the DRP it is being used as a reporter for, a western blot should be performed. As an alternative control, stop codons can be introduced between the repeat and Dendra2 to assure that all the signal is really representative of RAN translation.

4. In Figure 1D, the authors compare Dendra2 fluorescence in different cell types, and conclude that C9 RAN translation levels are higher in HEK293T cells, compared to NSC34 cells and rat cortical neurons. However, it is unclear if these differences are RAN-specific, or due to general differences in the levels or kinetics of plasmid expression across cell types, or even cell-type-specific differences in the ability of Dendra2 to fluoresce and be detected. Direct comparison of the fluorescence across cell types from a unmodified Dendra2 and mFP reporter could resolve this.

5. What reading frame is monitored in Figure 2A and S6?

6. In the text for Figure 4C, the authors overstate the increase observed in RAN translation with medium and high stimulation as ~2-fold.

7. In Figure 5A, it is unclear why the authors chose to use filter trap assays to measure the levels of soluble proteins, such as eIF2alpha and PERK, when western blots allow for better resolution and separate out any non-specific antibody interactions.

8. Regarding figure 5, what is this adding? If this is the first time someone has demonstrated eIF2alpha phosphorylation and PERK activation in the setting of excitotoxicity, then this finding should be highlighted. If this finding has been previously established or reported, then that work should be cited in the paper and this work becomes confirmatory of previous findings without significant importance here. Thus, this data could easily be removed as a main figure in the paper.

9. The text, figures, and legends for Figures 5A and S7, indicate that the authors use an anti-PERK antibody. However, in the methods, only an anti-phospho-PERK antibody is listed. This should be clarified, as phospho-PERK is a more definitive marker of ISR activation than increased PERK levels.

10. Fig 6. There is a very subtle overall reduction in the induction of RAN by stress. From the way the data is presented, it is masking the fact that the blockade by the inhibitors is incomplete- since it is normalized so many times as to make it hard to interpret. Moreover, I think the data un-normalized would actually show RAN going up, but not as significantly in the presence of the inhibitors. Please show the expression untreated, with TG or Na As, and then with TG or NaAs and the inhibitor all on the same scales so that it is clear what the effects of each is and the degree of inhibition. As stated above, I believe the claim in the abstract based on this data that these compounds "greatly inhibit RAN translation", is inaccurate.

11. A major claim of the paper (and something that is new compared to past work) is that they observe these stress-effects in neurons in particular. Yet these "rescue" studies are done without any actual rescue in NSC34 cells. Extension to neurons would really help the paper, and testing whether PERK This is especially true for Trazodone and 1,3 DBM. Some degree of phenotypic rescue in such neurons, which die with glutamate exposure, would really strengthen the manuscript.

12. The concentration of the compounds used in Figure 6 should be listed in the figure legend and/or methods.

13. Salubrinal inhibits the eIF2alpha phosphatase. Consequently, it increases phospho-eIF2alpha levels and activates downstream ISR pathways. Therefore, with the authors' data in Figure 5, it is not surprising that the authors do not observe a significant decrease in C9 RAN translation with salubrinal treatment in Figure 6. Also, the spelling of this compound should be checked throughout the text.

14. Supplement 1 and the antisense reporters- This represents (we think) the first report on reporters for antisense RAN translation from CCCCGG repeat expansions. As such, this data is more important than some that is included in the main paper and should be placed in the main text and more highlighted. Regarding these constructs, their exact surrounding sequence context is unclear. Is the AUG that is located normally in the PR frame above the repeat present in these constructs? Was this retained or eliminated? Some further details on these reporters would be of value.

Last point:

References to past work: This paper has some significant issues related to citation of past work in the field. There are numerous examples where the wrong paper or only one of multiple papers published simultaneously are cited for a finding or no paper is cited when data is presented that recapitulates published work. It is particularly egregious in the area of RAN translation mechanisms. To our knowledge, four papers have been published on mechanisms underlying RAN at GGGGCC repeats and three separate papers have been published on RAN mechanisms at other repeats. All need to be cited and all need to be referenced accordingly and accurately. Specific examples (not-exhaustive):

- Zu et al 2011 (referenced only for giving the name "RAN" translation) initially identified RAN at CAG and CUG repeats and showed differential effects in RRL and in different cell types, yet this paper is not even referenced when they present data in figure 1 that there are differences in C9 RAN translation in different systems.
- Todd et al 2013 and Kearse et al 2016 demonstrated initiation above or within the CGG repeat in different reading frames as well as cap and scanning dependence.
- Green et al 2017, Tabet et al, 2018 and Cheung et al, 2018 all demonstrated differential translation of GA>GP and GR, with Green et al and Tabet et al showing cap-dependence and some evidence for frameshifting while Cheung et al describing evidence for cap-independent initiation.
- Green et al 2017, Cheung et al, 2018 and Sonobe et al, 2018, *Neurobiology of Disease* (not referenced in this manuscript at all) all showed RAN induction with cellular stress, yet often only one or none of these papers is cited when discussing this.
- Cheung et al previously testing ISRIB and the GSK PERK inhibitor, yet data is presented in Figure 6 showing the GSK inhibition as though it was not previously published.

Please carefully review the previous literature, make sure that each statement correctly credits past work and make sure that findings presented as new are not a recapitulation of published results.

Referee #3 (Remarks for Author):

Westergard et al., established the cellular monitoring system of non-AUG (RAN) translation of G4C2 repeats in C9orf72 which is responsible for ALS/FTLD. The authors analyzed fluorescent signals by non-AUG (RAN) translation in different types of cells including NSC34 cells, rat primary cortical neurons, and human iPS-derived motor neurons harboring mutant G4C2 repeats in C9orf72. They found that various cellular stress or neuronal excitotoxic stress increases DPR levels for all C9orf72 NRE ORFs. The reporter fluorescent signals of DPR levels provoked by excitotoxic stress was reduced by AMPA or NMDA blockers. The authors manipulated neuronal excitation by optogenetics using Chr2 and found that neuronal activity increased non-AUG RAN translation of DPRs in neurons. Finally, they identified Perk-eIF2a-ATF4 pathway was involved in the increase in DPR translation caused by excitotoxicity.

The attempt to monitor non-AUG (RAN) translation of G4C2 repeats in C9orf72 is quite interesting and the findings that it can be driven by neuronal excitotoxicity and cellular stress are impactful. However, the monitoring system itself has many technical concerns, including inappropriate data presentation of the translated products of DPRs-Dendra2 proteins.

Furthermore, lack of in vivo evidence and unconvincing evaluation weaken the significance of the manuscript. Thus, the manuscript is not matured for the publication in EMM. The specific comments are listed below.

Major issues

1. The key construct of C9orf72-(G4C2)₁₈₈ NRE non-AUG-dependent reporter has many issues. First, the translated fluorescent protein should be DPR fused to Dendra2-HA. For instances, poly(GA)₁₈₈ corresponds to 24.42 kD whereas Dendra2 is 26 kD. The molecular behaviors in the cell can be different between 24.42 kD of poly(GA)₁₈₈ and 48.84 kD of poly(GA)₁₈₈-Dendra2-HA protein, although the authors insisted that the reporter protein mimic DPR pathological features in

Fig1. Second, the authors need to present not only filter trap assay but WB results of DPRs in Fig1C, Fig2B, and later. Otherwise, the quantification of "DPR levels" is not acceptable.

2. What would happen when not-in-frame with Dendra2-HA translation occur? It may yield DPR with a certain length of amino acids which is eventually stopped on the plasmid. It could cause non-specific phenomenon in the cell. The authors need to investigate and discuss upon this issue carefully.

3. Although the authors mentioned that there were measurable differences in sense DPR levels with the GA ORF predominantly the highest, the transfection efficiency among GA, GP, and GR could be different.

4. The data of Fig1C is not convincing that the reporter protein recapitulate DPR pathological features. The signals of Dendra2 look saturated and merged images with NES-mIFP are not clear. The authors should use Hoechst 33342 to stain the nucleus of living cells or use DAPI staining after fixation.

5. What is the criteria for "DPR positive" cells in Fig 1D and later?

6. Fig.2 and 3 are quite confusing and not adequate since the evaluated fluorescence is not the same among the cell types. For NSC34 and cortical neurons, the reporter Dendra2 signals were measured whereas the IHF using anti-DRP antibodies signals were used for iPSC-MNs from patients. The authors need to separate these figures.

7. It is strongly recommended to compare iPSC-derived MNs between patients and normal controls.

8. Thorough the manuscript the authors compared the biological differences of DPR translation among mouse NSC34 cells, rat primary cortical neurons, and human iPSC-derived motor neurons. This is technically helpful but is not biologically significant. It would be meaningful and impactful if the authors compare iPSC-derived MNs and iPSC-derived glial cells or other neuronal cell types.

9. In P8 L38 the authors mentioned that "excess neuronal calcium signaling may play a role in the production of DPRs through non-AUG-dependent translation". However, the rescue experiments using antagonists were always done under the agonist treatments. Such experiments only exclude the possibilities of non-specific phenomenon.

10. It is necessary to evaluate phosphorylated-Perk levels instead of total Perk in Fig.5. Moreover, the data of Fig 5A must be shown in WB with size markers. The quantification of protein levels by filter trap assays is not acceptable here.

11. It would be helpful if the authors show cell death/viability levels in Fig 5, 6, S6, and S7.

Minor issues

1. The authors need to discuss upon Fig. 2B in which Glu treatment did not increase DPR levels.

2. In P14 L6 "> 500" should be "m > 500".

3. In P16 L2 and 18 "Table2" should be "Table S2".

4. In Fig 5B, the graph of anti-ATF-4 is missing.

5. A typo in P3 L3, please correct the citation of Chesnokova:2017.

6. In the Table S2 legend, "see Figure S7" should be "see Figure S6".

Referee #1 (Remarks for Author):

Hexanucleotide repeat expansions of GGGGCC in the C9orf72 gene are the most common genetic cause of ALS and FTD. There are three hypotheses underlying potential disease mechanisms: 1) haploinsufficiency of C9orf72 protein, 2) gain of toxicity from sense and antisense RNA containing these repeats, and 3) non-canonical translation of these repeats leading to toxic dipeptide repeat species. Numerous studies have demonstrated that arginine-rich species (glycine-arginine and proline-arginine) are acutely cytotoxic and are therefore hypothesized to be the major contributors to disease. A major challenge in the field has been to uncover precisely how this non-canonical translation occurs and therapeutic avenues to prevent translation of these toxic dipeptide repeat species.

Westergard et al. recapitulate work from previous groups illustrating that the non-canonical translation of C9orf72 repeats is upregulated by various stressors using a novel dendra2 reporter system. This work supports the hypothesis that the integrated stress response (ISR) can upregulate non-canonical translation of C9orf72 repeats through phosphorylation of eIF2 α . Importantly, the authors provide two FDA approved therapeutics trazodone and 1,3 DBM that target eIF2 α phosphorylation.

A novel finding facilitated by their photo-convertible dendra2 reporter, was that the dipeptide repeat species are highly stable. In addition, the authors provide evidence that excitotoxic stress can also upregulate translation of C9orf72 repeats. Pharmacologically inhibiting AMPA or NMDA receptors was sufficient to reduce the levels of translation during excitotoxic stress, suggesting an importance for cationic influx as an early step to ISR activation.

Overall the work can be divided into 1) supporting previous work published by Green et al. 2017, Chang et al. 2018 and Sonobe et al. 2018 implicating cellular stress to upregulation of C9orf72 repeat translation and 2) additional work indicating that excitotoxic stress in neuronal cells upregulates C9orf72 repeat translation. This work collectively adds to our understanding of RAN translation mechanisms and is an important contribution. There are additional experiments and controls that should be considered prior to publication and are described in detail below.

Major concerns:

1) The authors do a nice job illustrating that despite which frame dendra2 is placed, all three potential dipeptide species can be detected (in order to illustrate that the addition of dendra2 does not influence non-canonical translation of other frames). They also show that dipeptide fusions recapitulate cellular localization of AUG-driven, codon-optimized DPRs. Based on this latter data, they claim that dendra2 is tagging specific DPRs.

a. Based off recent work from Tabet et al. (2018), there is evidence that frameshifting can occur with initiation starting at CUG, leading to additional +1 and +2 frame products. Given this data, indicating consistent localization is not sufficient to support the authors' conclusion. The proper experiment would be to do an HA-immunoprecipitation and subsequently probe for each of the three dipeptide species to confirm immunoreactivity with the DPR species being tagged. This would also help to provide more evidence for or against frameshifting.

Frameshifting has been shown to occur immediately upstream of repeats following the CUG cognate sequence. The Dendra2 reporter is located on the c-terminus of the DPR (downstream of the repeats) containing construct. Therefore, based on previous reports we should only be measuring one of the three frames with the Dendra 2, which is consistent with our new DPR colocalization experiments in Expanded View Figure EV1F. Our previous and new results also supports that frameshifting, at least upstream of the repeats, does occur; in Expanded View Figure EV1F we also demonstrate that all three frames are detected using DPR-specific antibodies regardless of Dendra2 frame. Together our new data provides evidence that the Dendra2 is reporting primarily for the designed frame/DPR since it robustly colocalizes with the expected DPR-specific antibodies, and frame shifting does occur consistent with previous findings as shown by our DPR blot dot analyses. Further investigation into frameshifting and mechanisms that alter frameshifting efficiency would require development of a new construct toolbox to rigorously examine this phenomenon.

b. Dot blots were only performed for all three DPRs when "ORF3" was tagged. Where is the data for the other two tagged frames? This should be included for completeness of the figure.

We have repeated the dot blot experiments on the other open reading frames and include them in Expanded View Figure EV1F.

2) DPR fluorescence quantifications & slot blots. The authors panel an impressive number of drugs in order to assay many different types of cellular stresses. However, in all quantifications of both fluorescence and slot blots there is no clear control that the levels of DPRs are compared against.

In the slot blot assays all samples were first normalized to an internal control (GAPDH) and then the change in RAN translation was shown relative to a non-stressed or DMSO-only treated cell (CTRL). We have updated the text and figure legends accordingly to make this clearer.

a. In the Figure S1 panel E, it is indicated that NES-mIFP is used as both a positive transfection marker and a readout of AUG-dependent translation. It is not clear from the method section, within figures, nor in figure legends whether or not the total level of DPRs has been normalized or even compared to AUG-dependent translation, although it is abundantly clear that NES-mIFP is being utilized as a transfection marker and cytosolic marker.

AUG-dependent translation was only used as a normalizer when comparing RAN translation efficiency among NES-mIFP positive cells for different cell types. This normalization was not used for subsequent analyses to minimize confounding factors that could differentially affect the ratio of AUG versus RAN translation. We have updated the text to improve clarity and address these issues.

i. To resolve this, the authors should include AUG-NES-mIFP (or another AUG-driven protein) expression levels in each experiment. Showing the levels of a control construct side-by-side to DPR levels would be more helpful than the normalization (see note ii below).

Please see above.

ii. The authors should make it clear in the methods section or figure legends that DPR levels have first been normalized to an AUG-dependent product and then graphed as fold-change relative to "ctrl" treated expression (If this is the case, which I'm assuming, but again not clear).

Please see above. Moreover, we have included in the figure legend that the AUG-NES-mIFP normalized DPR fluorescent intensity is represented as fold-change relative to HEK293T for comparisons among cell lines.

iii. It would also be ideal to have data on the mRNA levels of the reporter in each of the stressors to indicate that C9 reporter RNA increase does not account for the increase in DPR levels. At the very least, this must be included for the excitotoxic stress experiments.

We have now included reverse transcriptase quantitative PCR analyses that measure levels of NES-mIFP or Dendra2-containing transcripts in Expanded View Figure EV1G. These results indicate that the levels of DPRs measured in our study do not trend with transcript levels for the stressor experiments.

3) The authors suggest that calcium influx may be a signal that stimulates the ISR. Their conclusion is that "AMPA and NMDA channel antagonists can drastically reduce...excitotoxic stress-linked [translation]" and that excitotoxic stress can activate ISR as shown by increase in PERK and eIF2 α -P levels.

a. A good experiment that is missing here is to repeat the excitotoxic stress and inhibit eIF2 α phosphorylation or PERK activation (with drugs used in figure 7) and determine whether DPR levels are increased or reduced. If reduced consistent with the authors' model, this would help to really tie together the idea that excitotoxic stress through AMPA and NMDA receptors activates ISR. So far, I am not convinced the authors have provided sufficient data to conclude that excitotoxic stress converges on the ISR (suggested in text and figure 7).

As suggested, we now include in Figure 6 the results from testing therapeutic rescues in combination with glutamate induced excitotoxic stress in primary cortical neurons.

4) It is interesting that repetitive stimulation of primary neurons leads to an increase in DPR levels.

However, figure 4 needs work:

a. Why is only the GP frame data included here? What about at least the other two sense products, GA and GR?

In previous figures we have shown that the changes in RAN translation levels for all three frames are robustly comparable, and thus, can individually quantitatively reflect the overall changes in DPR levels. Therefore, we used the GP frame as a representative RAN translation since this provided the most diffuse and uniform distributed fluorescent signal.

b. This figure and authors conclusions "results indicate that repetitive neuronal depolarizations, which are possibly associated with increased neuronal activity promotes non-AUG translation" and further discussions give the overall sense that figure 4 is a weak link currently in the paper. The ALS field (as the authors point out the introduction) is decidedly at odds regarding whether hyper or hypoexcitability is occurring in ALS models. It is an interesting result, however I recommend moving it to the supplement if the experiment is not repeated for GA and GR (as recommend above).

We disagree with the reviewer comment on this point. The ALS field agrees that hyperexcitability occurs at some stage in most disease models. However, it is unclear the role or contribution that hyperexcitability may have on disease progression. In this work we make no argument that hyper- or hypo-excitability plays a direct role in neuronal death.

As stated in the response above, all three reading frames provide a similar readout for RAN translation and monitoring the GP frame accurately and quantitatively represents alterations to this general phenomenon for all three frames.

5) Figure 1D, how do the authors know that the differences in # of DPR positive cells and overall fluorescent intensity doesn't simply correspond to transfection efficiency (i.e. HEK293T cells are very well transfected). It's unclear again if only cells with NES-mIFP positive cells are counted, but even so the NES-mIFP reporter must be much smaller than the C9 reporters. For cells that are easy to transfect like HEKs, both reporters probably get taken up without much issue but you might find that the C9 reporters are harder to transfect into primary neurons.

We have modified the language throughout the text and in the figure legends to make it clearer that we are only measuring RAN translation levels in transfection positive cells, using the cotransfected NES-mIFP reporter.

We measured transcript levels using quantitative PCR and see that transcript levels do not directly correlate with differences in RAN translation levels (Expanded View Figure EV1G). Additionally, since cell-type differences in RAN translation is not the primary focus of this work but rather a noteworthy observation for those studying RAN translation for modeling disease, we have modified the text accordingly and intend to further investigate cell-type specific differences in iPSCs derived from patients carrying the C9orf72 in our future work. As a side note, the NES-mIFP reporter plasmid is almost double the size of the RAN translation reporter plasmid.

a. Also, CMV promoter is not the best for expression in neurons, so how can you compare expression in HEKs (where CMV works very well) to neuronal expression?

We have provided new results that examine the mRNA levels for constructs containing the RAN translation reporter versus the NES-mIFP reporter measured in both HEK293T and NSC34 cells indicate that increased RNA levels do not correlate well with increased RAN translation. Moreover, HEK293T cells show lower transcripts ratio for Dendra2/NES-mIFP but have higher RAN translation than NSC34 cells (Expanded View Figure EV1G). Therefore, we conclude that the differences in DPR levels between these cell lines are most likely due to changes in the efficiency for RAN translation and not necessarily changes in transcript levels. We have also updated the text as stated above.

i. Perhaps you can repeat with another promoter such as EF1a that work well in both HEK cells and neuronal cells. This also comes back to point 2iii, controlling for RNA expression.

Please see response above.

Minor points:

1. Introduction pg. 2-3: "One potential outcome....dynamic transcriptional regulation....P-eIF2 α dependent transcriptional regulation is a key response element to cellular stress."

We have corrected the text to say translational and not transcriptional.

a. Is the use transcription above a typo? Chesnokova et al. 2017 refer to translational changes caused by eIF2 α phosphorylation.

Please see immediately above.

2. Where are c9ALS patient-derived sMN data combined? It's well-established that different c9ALS lines express DPRs are different levels. It would be ideal in the supplement to see the lines separated.

Two C9 iPSC lines obtained from the same patient were used in this study to serve as endogenous RAN translation validation for our transient transfection findings in comparison to our overexpression models. Only the relative change in DPR levels upon stress were examined, and we did not compare DPR levels among different iPSC lines in this study. The line IDs from Target ALS are included in the Materials and Methods.

a. Unclear from IF methods whether DPR antibodies used in immunoblots are used again for IF (and if so, dilutions, etc. are missing).

The methods and figure legends were updated to improve procedural clarity.

3. Half-life calculations:

a. DPR half-life calculations don't even extend long enough for half the fluorescence to decrease. Is it really appropriate to calculate half-life without at least reaching this point?

We have updated the language in the text to say predicted half-life to be consistent with the table that shows the values for the calculated half-lives.

4. Overall microscopy images are very small and difficult to see (especially neuronal primary cells or patient cell data).

We apologize if the images appeared small and difficult to see and will modify the images to be consistent with publication standards as recommended by the editor.

a. Figure 1 and other cultured cells images are frequently oversaturated. It isn't appropriate to use oversaturated images for quantifications.

Oversaturated images were not used in quantification. The representative images are presented to improve visualization and publication appearances of fluorescent intensity.

5. Figure 2 A, please be clear which DPR you are using.

The GP frame was being used in Figure 2A, and we have updated the figure legend accordingly.

6. A small typo: Figure 3 A, "...the C9 DPR reporter andi in iPS...."

The typographical error has been corrected.

7. Figure S7 is a nice addition, is it possible to get the one-way tests for DPR levels as well?

a. This could be helpful for better understanding if a specific type of stress or class of stressors can be selective with regards to different ORFs.

All data was original assembled with one-way test comparisons being made for each DPR. However, we decided that comparing general RAN translation as whole (including all three DPRs together) would serve as a better representation of our findings and results. Understanding the subtle sensitivity difference of each DPR to different stressor treatments in combination with different cell types as well as exploring these relationships to different reading frames is outside of the scope of this work.

8. Should add Sonobe et al. 2018 to references of previously testing RAN translation and cellular stress (in addition to Greene et al. 2017 and Cheng et al. 2018).

The text has been updated to now include this new reference.

9. Figure 5B, is atf4 data missing here or is there an error in including it in the panel legend?

ATF4 has been removed from the figure legend in Figure 5B.

Referee #2 (Remarks for Author):

In the present manuscript, Westergard and colleagues assess the effect of neuronal excitation and cellular stress on RAN translation of 188 G4C2 repeats by expressing Dendra2-tagged DPR constructs in NSC34 cells, primary rat cortical neurons and C9 human iPSC-derived sMNs.

At the time of this review, to our knowledge, three other publications have shown the integrated stress response (ISR) to upregulate G4C2-RAN translation (Green et al., 2017, Cheng et al. 2018, and Sonobe et al., 2018). However, the present manuscript expands upon these previous publications in four key ways. 1) The researchers thoroughly assess the effects of a wider panel of cell stressors, including NMDA receptor mediated excitotoxicity, on C9 RAN translation than previous reports, implicating new stress pathways in this phenomenon. 2) Using C9 patient-derived iPS sMNs, they show that cell stress increases endogenous DPR levels. 3) They specifically establish a role for excitotoxicity in enhancing C9 RAN translation. 4) They identify two small molecules that reduce the increase in C9 RAN translation under conditions of exogenous cellular stress. Overall, these findings represent an advance from previous reports on how cell stress enhances C9 RAN translation, and of factors

that may promote disease in patients. A more significant advance could be potentially achieved with further studies on the newly identified suppressors of stress-induced C9 RAN translation (trazodone and 1,3DBM). has a few shortcomings that need to be addressed.

Concerns (in order of content):

1. In the abstract, the authors state that PERK inhibitors and other compounds "greatly reduce DPR levels." This statement is not supported by the data in the paper. In figure 6, there is a relative reduction in ability of stress to enhance DPR reporter expression, but this is likely not a reduction at all (it is not possible to interpret from the multiply normalized data in this figure) and it is certainly not a "great" reduction.

We have modified the language in the abstract to better represent the extent of rescue observed here. Additionally, Figure 6 has been updated to show the pharmacological reduction of DPR levels relative to unstressed controls and not to stressed controls as previously presented. However, our statistical comparisons are still performed to measure the significance of the pharmacological rescue from these stressed conditions.

2. On pages 2-3, in the second to last paragraph of the introduction, the authors should clarify that altered translation is a direct consequence of eIF2 α phosphorylation, with transcriptional changes occurring as a result of altered translation of transcription factors such as ATF4.

There was an error in the paragraph highlighted by this comment, "transcriptional" was incorrectly used in place of "translational". This paragraph was meant to highlight only the translational changes attributed to p-eif2 α . The downstream changes and role of ATF4 are highlighted in the results section regarding ISR protein levels.

3. In Figure 1B-C, the authors use filter trap assays with an anti-HA antibody and Dendra2 fluorescence as readouts for GA, GP, and GR RAN translation from reporter constructs. However, it is possible that Dendra2-HA is expressed independent of RAN translation. To clearly show that Dendra2-HA is fused to the DPR it is being used as a reporter for, a western blot should be performed. As an alternative control, stop codons can be introduced between the repeat and Dendra2 to assure that all the signal is really representative of RAN translation.

We have included in Expanded View Figure EV1F immunofluorescent colocalization experiments using antibodies specific for DPRs and HA. In this updated figure we show that all three DPRs are generated for each construct, but the DPR-Dendra2-HA fusion protein colocalizes perfectly with the DPR in the Dendra2-HA open reading frame, as shown by the fluorescent intensity overlap of the DPR-specific antibody with the DPR-Dendra2-HA fusion protein.

4. In Figure 1D, the authors compare Dendra2 fluorescence in different cell types, and conclude that C9 RAN translation levels are higher in HEK293T cells, compared to NSC34 cells and rat cortical neurons. However, it is unclear if these differences are RAN-specific, or due to general differences in the levels or kinetics of plasmid expression across cell types, or even cell-type-specific differences in the ability of Dendra2 to fluoresce and be detected. Direct comparison of the fluorescence across cell types from a unmodified Dendra2 and mIFP reporter could resolve this.

New data examining the mRNA levels for constructs containing the RAN translation reporter versus the NES-mIFP reporter measured in both HEK293T and NSC34 cells indicate that increased RNA levels do not correlate with increased RAN translation – HEK293T show lower transcript levels but higher RAN translation than NSC34 cells, as shown in Expanded View Figure EV1G. Thus, the differences in DPR levels between these cell lines are driven by changes in RAN translation efficiency. Cell-type-specific difference for the quantum yield of Dendra2 or even NES-mIFP at these emission and excitation wavelengths would be more sensitive to the relative intensity of Dendra2 compared to NES-mIFP when normalized as in Figure 1D. However, we also demonstrate that the number of cells positive for RAN translation in transfection positive cells, which should be less sensitive to changes in quantum yields, provides similar results. Therefore, the quantum yields of Dendra2 would need to be drastically altered to explain these similar observations. Additionally, we have not found any reports examining the biophysical characterizations of Dendra2 in different cell-types or suggesting there could be differences. Additional biophysical characterizations for changes to quantum yield of Dendra2 in different cell types is outside of the scope of this study.

5. What reading frame is monitored in Figure 2A and S6?

We have updated the figure legends to clarify that we are utilizing the C9 DPR reporter in the GP frame in these experiments.

6. In the text for Figure 4C, the authors overstate the increase observed in RAN translation with medium and high stimulation as ~2-fold.

We have corrected the text with the precise fold change.

7. In Figure 5A, it is unclear why the authors chose to use filter trap assays to measure the levels of soluble proteins, such as eIF2 α and PERK, when western blots allow for better resolution and separate out any non-specific antibody interactions.

The original blots were obtained through filter trap assays for parallel comparisons of DPR levels, using DPR-specific antibodies, to changes in ISR proteins. Filter trap assays are typically used to measure DPRs due to the relatively low levels of DPRs and difficulties in analyzing DPRs in PAGE analyses. However, we now include new western blot results in Expanded View Figure EV7C that show similar trends to our filter trap assays. This new result further validates our original findings and that the usage of the filter trap assay can accurately represent changes in the levels for ISR proteins using these antibodies.

8. Regarding figure 5, what is this adding? If this is the first time someone has demonstrated eIF2 α phosphorylation and PERK activation in the setting of excitotoxicity, then this finding should be highlighted. If this finding has been previously established or reported, then that work

should be cited in the paper and this work becomes confirmatory of previous findings without significant importance here. Thus, this data could easily be removed as a main figure in the paper.

The activation of ISR related pathways through excitotoxic stress in neurons is still something that not fully understood. There is much data indicating that in neurons, p-eIF2 α -driven translational shift is a necessary part of normal neuronal function in the absence of stress. Additionally, these pathways are implicated in establishing LTP and LTD, particularly through glutamatergic stimulation of GluN2B translation (Chesnokova:2017dd). The focus of Figure 5 is to further understand how the ISR pathway responds to excitotoxic stress, how preventing certain receptor activation modulates the pathway, and how this correlates to RAN translation.

9. The text, figures, and legends for Figures 5A and S7, indicate that the authors use an anti-PERK antibody. However, in the methods, only an anti-phospho-PERK antibody is listed. This should be clarified, as phospho-PERK is a more definitive marker of ISR activation than increased PERK levels.

We have corrected the figures and figure legends to correctly represent the results examining phospho-PERK levels.

10. Fig 6. There is a very subtle overall reduction in the induction of RAN by stress. From the way the data is presented, it is masking the fact that the blockade by the inhibitors is incomplete- since it is normalized so many times as to make it hard to interpret. Moreover, I think the data un-normalized would actually show RAN going up, but not as significantly in the presence of the inhibitors. Please show the expression untreated, with TG or Na As, and then with TG or NaAs and the inhibitor all on the same scales so that it is clear what the effects of each is and the degree of inhibition. As stated above, I believe the claim in the abstract based on this data that these compounds "greatly inhibit RAN translation" is inaccurate .

We have updated this figure to include the untreated cells with the treated + inhibitors. Indeed, the inhibitors do not reduce RAN translation levels back to basal levels but do significantly reduce the effect of stressors on RAN translation. We have modified the text to more accurately reflect the extent of pharmacological rescue we observe. However, we believe this to be an important finding as these inhibitors are significantly reducing RAN translation levels under constant stress, while cellular stress occurs at varying intervals in individuals. These therapies can therefore serve to stifle the levels of RAN translation during higher levels of cellular stress.

11. A major claim of the paper (and something that is new compared to past work) is that they observe these stress-effects in neurons in particular. Yet these "rescue" studies are done without any actual rescue in NSC34 cells. Extension to neurons would really help the paper, and testing whether PERK This is especially true for Trazodone and 1,3 DBM. Some degree of phenotypic rescue in such neurons, which die with glutamate exposure, would really strengthen the manuscript.

We now include in Figure 6 the results from testing the possible therapeutic rescues in combination with glutamate induced excitotoxic stress in primary cortical neurons.

12. The concentration of the compounds used in Figure 6 should be listed in the figure legend and/or methods.

We have updated the methods and figure legend in Figure 6 to include the concentrations of all compounds used.

13. Salubrinal inhibits the eIF2 α phosphatase. Consequently, it increases phospho-eIF2 α levels and activates downstream ISR pathways. Therefore, with the authors' data in Figure 5, it is not surprising that the authors do not observe a significant decrease in C9 RAN translation with salubrinal treatment in Figure 6. Also, the spelling of this compound should be checked throughout the text.

Our approach in finding inhibitors of RAN translation was to use small molecule inhibitors that target the ISR or translation-related pathways that are currently in use or have been used in clinical trials for neurodegenerative diseases or cancer. Salubrinal was one of these compounds. The

reviewer is correct, Salubrinal prevents the dephosphorylation of P-eif2 α , however, inclusion of this small molecule was important to further resolve the role of p-eif2 α in RAN translation and therapeutic intervention strategies. We have corrected the spelling throughout the text.

14. Supplement 1 and the antisense reporters- This represents (we think) the first report on reporters for antisense RAN translation from CCGG repeat expansions. As such, this data is more important than some that is included in the main paper and should be placed in the main text and more highlighted. Regarding these constructs, their exact surrounding sequence context is unclear. Is the AUG that is located normally in the PR frame above the repeat present in these constructs? Was this retained or eliminated? Some further details on these reporters would be of value.

The legend text for Expanded View Figure EV1 has been modified to more clearly assert that same sense sequence construct was used with only the repeats being flipped relative to the flanking sequences. The antisense promoter region is not as clearly defined as the sense promoter region for the C9orf72 NRE. Therefore, we chose not to presume where the transcriptional start site for the upstream antisense sequences originated, but instead focused on the half-lives of these antisense DPRs relative to the sense DPRs under similar conditions. Due to the antisense DPRs not having the undefined endogenous upstream regions required for RAN translation of all three reading frames, we were cautious to highlight these findings beyond the context of DPR calculated half-lives as presented in the Expanded View figures.

Last point:

References to past work: This paper has some significant issues related to citation of past work in the field. There are numerous examples where the wrong paper or only one of multiple papers published simultaneously are cited for a finding or no paper is cited when data is presented that recapitulates published work. It is particularly egregious in the area of RAN translation mechanisms. To our knowledge, four papers have been published on mechanisms underlying RAN at GGGGCC repeats and three separate papers have been published on RAN mechanisms at other repeats. All need to be cited and all need to be referenced accordingly and accurately. Specific examples (not-exhaustive):

- Zu et al 2011 (referenced only for giving the name "RAN" translation) initially identified RAN at CAG and CUG repeats and showed differential effects in RRL and in different cell types, yet this paper is not even referenced when they present data in figure 1 that there are differences in C9 RAN translation in different systems.
- Todd et al 2013 and Kearse et al 2016 demonstrated initiation above or within the CGG repeat in different reading frames as well as cap and scanning dependence.
- Green et al 2017, Tabet et al, 2018 and Cheung et al, 2018 all demonstrated differential translation of GA>GP and GR, with Green et al and Tabet et al showing cap-dependence and some evidence for frameshifting while Cheung et al describing evidence for cap-independent initiation.
- Green et al 2017, Cheung et al, 2018 and Sonobe et al, 2018, *Neurobiology of Disease* (not referenced in this manuscript at all) all showed RAN induction with cellular stress, yet often only one or none of these papers is cited when discussing this.
- Cheung et al previously testing ISRIB and the GSK PERK inhibitor, yet data is presented in Figure 6 showing the GSK inhibition as though it was not previously published.

We have now updated the text to include these references relevant to this work as suggested.

Please carefully review the previous literature, make sure that each statement correctly credits past work and make sure that findings presented as new are not a recapitulation of published results.

The Introduction, Results, and Discussion have been updated to appropriately include past work relevant to this study.

Referee #3 (Remarks for Author):

Westergard et al., established the cellular monitoring system of non-AUG (RAN) translation of G4C2 repeats in C9orf72 which is responsible for ALS/FTLD. The authors analyzed fluorescent signals by non-AUG (RAN) translation in different types of cells including NSC34 cells, rat primary cortical neurons, and human iPS-derived motor neurons harboring mutant G4C2 repeats in C9orf72.

They found that various cellular stress or neuronal excitotoxic stress increases DPR levels for all C9orf72 NRE ORFs. The reporter fluorescent signals of DPR levels provoked by excitotoxic stress was reduced by AMPA or NMDA blockers. The authors manipulated neuronal excitation by optogenetics using ChR2 and found that neuronal activity increased non-AUG RAN translation of DPRs in neurons. Finally, they identified Perk-eIF2a-ATF4 pathway was involved in the increase in DPR translation caused by excitotoxicity.

The attempt to monitor non-AUG (RAN) translation of G4C2 repeats in C9orf72 is quite interesting and the findings that it can be driven by neuronal excitotoxicity and cellular stress are impactful.

However, the monitoring system itself has many technical concerns, including inappropriate data presentation of the translated products of DPRs-Dendra2 proteins.

Furthermore, lack of *in vivo* evidence and unconvincing evaluation weaken the significance of the manuscript. Thus, the manuscript is not matured for the publication in EMM. The specific comments are listed below.

Major issues

1. The key construct of C9orf72-(G4C2)188 NRE non-AUG-dependent reporter has many issues. First, the translated fluorescent protein should be DPR fused to Dendra2-HA. For instances, poly(GA)188 corresponds to 24.42 kD whereas Dendra2 is 26 kD. The molecular behaviors in the cell can be different between 24.42 kD of poly(GA)188 and 48.84 kD of poly(GA)188-Dendra2-HA protein, although the authors insisted that the reporter protein mimic DPR pathological features in Fig1. Second, the authors need to present not only filter trap assay but WB results of DPRs in Fig1C, Fig2B, and later. Otherwise, the quantification of "DPR levels" is not acceptable.

The DPR fusion reporter construct was generated to primarily observe DPR production through RAN translation and to better understand cellular mechanisms that drive RAN translation. It was not developed for pathological recapitulation. However, it is intriguing that even with the fusion modifications we see many similar phenotypical features to other DPR-fusion proteins and often to DPR pathology in patient tissues. Moreover, the DPRs produced in this way have consistent localization patterns with both 1) previously used AUG-driven fluorescently tagged DPR constructs that have been utilized in numerous papers to understand DPR pathology and 2) frequently mirror pathological observations in patient-derived iPS neurons and post-mortem tissue. Although we agree with the reviewer that the inclusion of a fluorescent protein tag can alter the intrinsic properties of the protein of interest, in Wen et al., 2014, we demonstrated that fluorescently untagged DPR constructs have similar localization patterns and toxicity associated with them.

Filter trap assays are a commonly employed procedure for detecting DPRs as they are difficult to analyze on PAGE followed by western blot analyses due to: the low abundance of RAN translation products in transfection positive cells, the unusual aggregation in the stacking, the smearing of bands, and/or the DPR migration mobility being inconsistent with the molecular weight. We provided more than one analyses of our DPR reporter construct, and we observe consistent results among these analyses.

2. What would happen when not-in-frame with Dendra2-HA translation occur? It may yield DPR with a certain length of amino acids which is eventually stopped on the plasmid. It could cause non-specific phenomenon in the cell. The authors need to investigate and discuss upon this issue carefully.

Others have recently demonstrated that frame shifting does occur upstream of the repeats, and our result demonstrate that DPRs in all three reading frames are being produced (i.e. the GA-ORF reporter construct also produces GP and GR). Stop codons are located within the first 10 amino acids in the other two frames of Dendra2. With this construct we are not in a position to address frameshifting but have thoughtfully designed this construct to minimize potential deleterious frame-shift products that would produce long c-termini. Complete plasmid map DNA files included as supplemental material in the final submission will also address these concerns.

3. Although the authors mentioned that there were measurable differences in sense DPR levels with the GA ORF predominantly the highest, the transfection efficiency among GA, GP, and GR could be different.

New results are included in Expanded View Figure EV1G, showing that transcript levels measured using quantitative PCR do not directly correlate with or explain the differences in DPR levels. Furthermore, these constructs only vary by one or two base pairs, which should have a negligible effect on transfection efficiency. Additionally, multiple rounds of plasmids were isolated and purified to perform these experiments, which should offset any batch-specific differences in transfection efficiency for plasmid preparations with different ratios of supercoiled versus circular/nicked plasmid DNA.

4. The data of Fig1C is not convincing that the reporter protein recapitulate DPR pathological features. The signals of Dendra2 look saturated and merged images with NES-mIFP are not clear. The authors should use Hoechst 33342 to stain the nucleus of living cells or use DAPI staining after fixation.

While we disagree with the reviewer that the localization patterns are not consistent with what has been previously shown, the focus of this paper is not on recapitulating DPR pathological features, but primarily on drivers and inhibitors of RAN translation in the context of the *C9orf72* NRE. The representative images are for ease of visualization regardless of how the media is being presented.

5. What is the criteria for "DPR positive" cells in Fig 1D and later?

We have made it clearer throughout the text and figure legends that we are only measuring RAN translation levels in transfection positive cells, using the cotransfected NES-mIFP fluorescent reporter.

6. Fig.2 and 3 are quite confusing and not adequate since the evaluated fluorescence is not the same among the cell types. For NSC34 and cortical neurons, the reporter Dendra2 signals were measured whereas the IHF using anti-DRP antibodies signals were used for iPSC-MNs from patients. The authors need to separate these figures.

In Figure 2 and 3 we are not comparing between cell types but are measuring changes in DPRs levels in cell types in response to stressors. Figure 2 addresses cellular stressor and the response to these stressors in a number of cells, while Figure 3 focuses on excitotoxic stress. In Figure 2 we have now separated the iPSC figure to improve clarity for different analyses being performed (now Figure 2D).

7. It is strongly recommended to compare iPSC-derived MNs between patients and normal controls.

A major focus of this paper is to understand drivers and inhibitors of RAN translation by measuring changes in DPR production. Normal patient-derived iPSC controls should not contain the *C9orf72* NRE and therefore not produce DPRs – they show little to no DPR-positive staining.

8. Throughout the manuscript the authors compared the biological differences of DPR translation among mouse NSC34 cells, rat primary cortical neurons, and human iPSC-derived motor neurons. This is technically helpful but is not biologically significant. It would be meaningful and impactful if the authors compare iPSC-derived MNs and iPSC-derived glial cells or other neuronal cell types.

Cell-type specific differences are not the primary focus of this work, but rather an interesting finding that may be of importance to those studying disease mechanisms of RAN translation, and we have updated the text accordingly. We agree that understanding RAN translation in cell types derived from patient iPSC would be a meaningful and beneficial study and we absolutely plan to explore this avenue and *in vivo* models in future studies.

9. In P8 L38 the authors mentioned that "excess neuronal calcium signaling may play a role in the production of DPRs through non-AUG-dependent translation". However, the rescue experiments using antagonists were always done under the agonist treatments. Such experiments only exclude the possibilities of non-specific phenomenon.

We suggest that excess neuronal calcium signaling may play a role in the production of DPRs with our findings, but we mention in our discussion (P12 Para 2) that we don't discount the role

metabotropic receptors may have in the signalling cascades for RAN translation and that this will be a focus of future studies.

10. It is necessary to evaluate phosphorylated-Perk levels instead of total Perk in Fig.5. Moreover, the data of Fig 5A must be shown in WB with size markers. The quantification of protein levels by filter trap assays is not acceptable here.

We corrected the figures and figure legend to show that we analyzed phosphorylated-PERK. We have also included new western blots probing for the ISR proteins shown in Expanded View Figure EV7C. These findings are consistent with our filter trap assays and serve to validate our findings and usage of the filter trap assay to measure ISR proteins using these antibodies.

11. It would be helpful if the authors show cell death/viability levels in Fig 5, 6, S6, and S7.

Implementing cell death/viability throughout the paper was an initial goal of ours, but there are multiple confounding factors that precluded this type of analyses. For example, this would require longitudinal single-cell imaging and correlating the DPR levels with risk of death for DPR-linked toxicity, with the effect each stressor had on increasing DPR levels, with the relative intrinsic toxicity of the stressor, and with the effects on AUG-translation (NES-mIFP). While these multidimensional longitudinal analyses would be incredibly interesting, it would be extremely difficult and complicated to perform these experiments to properly ascribe the true nature of cell death.

Minor issues

1. The authors need to discuss upon Fig. 2B in which Glu treatment did not increase DPR levels.

Fig 2B shows the densitometric change in DPR levels that are measured using DPR-specific antibodies for total cells, which includes untransfected cells, transfected cells, and transfected cells producing DPRs and transfected cells not producing DPRs. Therefore, this method is typically less sensitive when normalized to total cell GAPDH since this normalization is diluting the signal from the cells that are producing DPRs in transiently transfected cells. While the fluorescent quantification is a single cell approach that measures DPR-Dendra2 levels only in transfection positive. We do address that glutamate has a more significant role in primary and iPSC neurons, which becomes the main focus of the following results.

2. In P14 L6 "> 500" should be "m > 500".

We have updated the text.

3. In P16 L2 and 18 "Table2" should be "Table S2".

We have updated the text.

4. In Fig 5B, the graph of anti-ATF-4 is missing.

This was an error and the anti-ATF4 label has been removed

5. A typo in P3 L3, please correct the citation of Chesnokova:2017.

We have updated the text.

6. In the Table S2 legend, "see Figure S7" should be "see Figure S6".

We have updated the text.

Thank you for the submission of your revised manuscript to EMBO Molecular Medicine. We have now received the enclosed reports from the referees that were asked to re-assess it. As you will see

the reviewers are globally supportive and I am pleased to inform you that we will be able to accept your manuscript pending the following final amendments:

1) Please address the comments of all referees. It is important to update the literature and amend figures as suggested, as well as text. Please note the request of referee 3. You can submit the data as source data if you prefer.

Please submit your revised manuscript within two weeks.

I look forward to reading a new revised version of your manuscript as soon as possible.

***** Reviewer's comments *****

Referee #1 (Comments on Novelty/Model System for Author):

experiments and model systems are appropriate

Referee #1 (Remarks for Author):

I think that this paper is now suitable for publication. The authors really did a thorough job going through almost all of our comments with either experiments or clarifications to the text, methods or figure legends. It seems like they've addressed many of the other reviewers' comments as well. We just have 2 minor but irksome points that we think the authors and editors should see and consider addressing:

Westergard et al. recapitulate work from previous groups illustrating that the non-canonical translation of C9orf72 repeats is upregulated by various stressors using a novel dendra2 reporter system. This work supports the hypothesis that the integrated stress response (ISR) can upregulate non-canonical translation of C9orf72 repeats through phosphorylation of eIF2 α . Importantly, the authors provide two FDA approved therapeutics trazodone and 1,3 DBM that target eIF2 α phosphorylation.

As stated previously, this work collectively adds to our understanding of RAN translation mechanisms and is an important contribution. Many of our comments and suggested experiments have been addressed in the resubmission and the manuscript will contribute to the field. We provide a few comments for consideration to the authors and editor prior to the final markup of the work.

Comments:

- C9orf72 reporter in Figure 1: Why not just replace n with 188 in panels A and B??
- "Our previous and new results also supports that frameshifting, at least upstream of the repeats, does occur; in Expanded View Figure EV1F we also demonstrate that all three frames are detected using DPR-specific antibodies regardless of Dendra2 frame. Together our new data provides evidence that the Dendra2 is reporting primarily for the designed frame/DPR since it robustly colocalizes with the expected DPR-specific antibodies, and frame shifting does occur consistent with previous findings as shown by our DPR blot dot analyses. Further investigation into frameshifting and mechanisms that alter frameshifting efficiency would require development of a new construct toolbox to rigorously examine this phenomenon."
 - o We are seriously concerned that Westergard et al. may have a misunderstanding of how to experimentally support the phenomenon of frameshifting. The data indicated above does not support frameshifting in any sense-it merely illustrates that regardless of which frame is tagged, RAN translation can still occur in all three frames. Key experiments to support frameshifting include disrupting the start site of one frame, and illustrating effects on other frames (conducted by Tabet et al. 2018) or mass spec analyses that clearly show hybrid species from two ORFs.
- "In previous figures we have shown that the changes in RAN translation levels for all three frames are robustly comparable, and thus, can individually quantitatively reflect the overall changes in DPR levels. Therefore, we used the GP frame as a representative RAN translation since this provided the most diffuse and uniform distributed fluorescent signal." & "Given robustly comparable RAN translation levels for all three frames of the C9 reporter construct in the presence of stressors, this

and subsequent experiments utilized the GP frame C9 reporter which provides diffuse, uniform fluorescent signal for quantification." Is added to text in response to our comments.

o We thank the authors for including clarity in the text as cited above. The author makes a good argument for studying diffuse GP as a readout for RAN translation. However, it is still inappropriate and misleading throughout the rest of the text and figure titles and legends concerning figures 4, 5, and 6 to refer generally to all DPRs or even ambiguously DPR. In the titles at the very least, it should clearly say the reduction or increase is occurring in polyGP and within each figure legend of 5 and 6 it should be added (not just to figure 4). This should be updated prior to publication-please consider the reality that many readers will go directly to the data/figures and skim over your text.

Referee #2 (Remarks for Author):

The reviewers have addressed my major concerns. The following changes would be appropriate prior to publication.

Minor issues:

- 1) In introduction, the following reference should be added at the end of the first paragraph: "the production of polypeptides through the unconventional nonAUG-dependent translation of the NRE region, frequently referred to as repeat-associated nonAUG initiated (RAN) translation (Mori et al., 2013a; Zu et al., 2013; 2011; Ash et al, 2013).
- 2) In introduction, the sentence below should be corrected as laid out to match existing nomenclature and to reference the correct work. "Recent mechanistic exploration of this translational phenomenon for the (G4C2)_n sense transcript has suggests that the non-AUG-dependent translation initiates predominantly from a near cognate AUG codon near cognate CUG codon upstream of the repeat (Kearse et al., 2016; Tabet et al., 2018; Todd et al., 2013 Green et al, 2017)"
- 3) In introduction, the following sentences would be more correct if they were changed to the following: "One potential outcome driver of from these events is the dynamic translational regulation of mediated by phosphorylation of the α -subunit of eukaryotic translation initiation factor 2 (p-eif2 α). Peif2 α is phosphorylated in response to cellular stress, which suppresses global protein translational initiation. However, this same phosphorylation event favors non-canonical translation initiation events on mRNAs (Chesnokova et al., 2017), and is implicated to have a role in both CGG and C9orf72 NRE-linked RAN translation (Cheng et al., 2018; Green et al., 2017)
- 4) In results, change sentence "Given robustly roughly comparable RAN translation levels for all three frames of the C9 reporter construct in the presence of stressors, this and subsequent experiments utilized the GP frame C9 reporter which provides diffuse, uniform fluorescent signal for quantification."

Referee #3 (Remarks for Author):

The authors partially improved the manuscript by responding some issues raised by the reviewer; however, many issues still remained unaddressed.

1. It has been know that many of poly dipeptides are easily smeared in PAGE gel except for poly (GA). For instances, Lee et al., demonstrated poly (GA) in WB (Hum Mol Gene, 2017). Zhang et al., also presented WB data of poly (GA) (Nat Neurosci, 2016). In this manuscript, the authors used artificial DRPs fused to Dendra2-HA; therefore, it is necessary to show the data of WB no matter how they look like.
2. The authors are in a position to address frameshifting by comparing the expression levels of translated products by WB.
4. Fig 1C remains quite unreliable.
10. The data of Fig 5A must be shown in WB with size markers instead of filter trap assays as mentioned in the first round review comments. This was also requested by the other reviewer. Figure EV7C showed that there was almost no signals of P-Perk in the control lane which was not consistent with that in the filter trap assay in Fig 5.

We would like to thank you and the three reviewers for taking the time to provide us with additional feedback, thoughtful suggestions, and the opportunity to improve our manuscript to make it more suitable for publication in *EMBO Molecular Medicine*. We have provided a point-by-point responses to address requests or suggestions raised by you and/or the three reviewers immediately below.

1) Please address the comments of all referees. It is important to update the literature and amend figures as suggested, as well as text. Please note the request of referee 3. You can submit the data as source data if you prefer.

In this second amended version of our manuscript, we have addressed all the additional comments provided both by the editor and the reviewers. With regard to the request of reviewer 3 to study RAN-translation frame shifting events that occurs in cells under different conditions, we strongly believe that although mechanistically interesting and relevant to C9orf72-linked pathogenesis, this line of investigation is outside the scope of the current manuscript. It is true – as the reviewer stated – that we have a tool that allows us to perform these new experiments. However, the timeline of the suggested new research direction and the complexity of the questions raised by the reviewer are such that they require a devoted project with better tools, manpower and time, considerably more time than the timeline allowed by the editor to resubmit the second revised version of this manuscript (see also our response to the comment of reviewer #1 on the issue of frame-shifting). In addition, as also indicated by reviewers #1 and #2, we believe that the data presented in this manuscript collectively already add many novel insights to our current understanding of RAN translation mechanisms and that the manuscript in its current form is an important contribution to the field.

Referee #1 (Remarks for Author):

I think that this paper is now suitable for publication. The authors really did a thorough job going through almost all of our comments with either experiments or clarifications to the text, methods or figure legends. It seems like they've addressed many of the other reviewers' comments as well. We just have 2 minor but irksome points that we think the authors and editors should see and consider addressing:

Westergard et al. recapitulate work from previous groups illustrating that the non-canonical translation of C9orf72 repeats is upregulated by various stressors using a novel dendra2 reporter system. This work supports the hypothesis that the integrated stress response (ISR) can upregulate non-canonical translation of C9orf72 repeats through phosphorylation of eIF2 α . Importantly, the authors provide two FDA approved therapeutics trazodone and 1,3 DBM that target eIF2 α phosphorylation.

As stated previously, this work collectively adds to our understanding of RAN translation mechanisms and is an important contribution. Many of our comments and suggested experiments have been addressed in the resubmission and the manuscript will contribute to the field. We provide a few comments for consideration to the authors and editor prior to the final markup of the work.

Comments:

- C9orf72 reporter in Figure 1: Why not just replace n with 188 in panels A and B??

We have updated the figure to show 188 repeats, which were used throughout this work.

- "Our previous and new results also supports that frameshifting, at least upstream of the repeats, does occur; in Expanded View Figure EV1F we also demonstrate that all three frames are detected using DPR-specific antibodies regardless of Dendra2 frame. Together our new data provides

evidence that the Dendra2 is reporting primarily for the designed frame/DPR since it robustly colocalizes with the expected DPR-specific antibodies, and frame shifting does occur consistent with previous findings as shown by our DPR blot dot analyses. Further investigation into frameshifting and mechanisms that alter frameshifting efficiency would require development of a new construct toolbox to rigorously examine this phenomenon."

o We are seriously concerned that Westergard et al. may have a misunderstanding of how to experimentally support the phenomenon of frameshifting. The data indicated above does not support frameshifting in any sense-it merely illustrates that regardless of which frame is tagged, RAN translation can still occur in all three frames. Key experiments to support frameshifting include disrupting the start site of one frame, and illustrating effects on other frames (conducted by Tabet et al. 2018) or mass spec analyses that clearly show hybrid species from two ORFs.

The reviewer is correct, in the work reported here we did not modify the near-cognate start codon upstream of the repeats to assess how this effects translation of all three reading frames as in Tabet et al. 2018. We appreciate the reviewers' interest in frameshifting because we are investigating frameshifting during translational elongation. However, during this preliminary investigation we came to recognize the need to develop more appropriate tools to quantitatively measure frameshifting. Furthermore, we have been in touch with Mass Spec facilities, and they have been reluctant to initially work on DPR frameshifting. This is primarily due to developing alternative experimental workflows to analyze peptide fragments that may have increased trypsin sensitive sites for any peptides that contain the arginine-rich dipeptides. The analyses of trypsin-digested fragments is routine in Mass Spec facilities compared to analyses of peptide fragments generated from employing chymotrypsin and/or other proteases to generate peptidal fragments. Furthermore, if we employed canonical trypsin digestion techniques it would be difficult to ascribe if or when shifting occurs out or into a arginine-rich dipeptide frame if that is due to: 1) translation initiated within the repeat and extended into the c-terminal fusion protein, 2) ribosomal pausing occurred within the repeats producing a truncated peptide fragment, 3) or a combination of these confounding and possibly other confounding events using our current construct. However, we are developing constructs to quantitatively measure frameshifting during translational elongation in bulk and single-molecule in vitro analyses. This will provide a more rigorous mechanistic understanding of frameshifting that is necessary to perform prior to presenting frameshifting to the scientific community, which is beyond the scope of the work presented here.

• "In previous figures we have shown that the changes in RAN translation levels for all three frames are robustly comparable, and thus, can individually quantitatively reflect the overall changes in DPR levels. Therefore, we used the GP frame as a representative RAN translation since this provided the most diffuse and uniform distributed fluorescent signal." & "Given robustly comparable RAN translation levels for all three frames of the C9 reporter construct in the presence of stressors, this and subsequent experiments utilized the GP frame C9 reporter which provides diffuse, uniform fluorescent signal for quantification." Is added to text in response to our comments.

o We thank the authors for including clarity in the text as cited above. The author makes a good argument for studying diffuse GP as a readout for RAN translation. However, it is still inappropriate and misleading throughout the rest of the text and figure titles and legends concerning figures 4, 5, and 6 to refer generally to all DPRs or even ambiguously DPR. In the titles at the very least, it should clearly say the reduction or increase is occurring in polyGP and within each figure legend of 5 and 6 it should be added (not just to figure 4). This should be updated prior to publication-please consider the reality that many readers will go directly to the data/figures and skim over your text.

We disagree with the reviewer on this point. We are observing alterations in RAN translation in the context of C9orf72 via DPR production. As mentioned in our previous response, we thoroughly demonstrate that changes in RAN translation affect DPR production regardless of what DPR is being made. Figure 4 (the first figure mentioned by the reviewer) is an extended experimental design based on Figure 3 in which we utilized an in-depth approach to observe all sense DPR changes in the context of excitotoxic stress, both in primary cortical neurons and human derived iPSC. We already validated all DPR changes in Figure 3 and to extend Figure 4 into all DPR

production would be a costly and timely addition to an already difficult approach. Figure 5 (the second figure mentioned by the reviewer) contains no analysis of DPR levels, but solely focuses on the ISR related proteins and the changes associated with them. DPR production is only used as a marker for cells undergoing RAN translation so that we can measure the ISR proteins within those cells. Figure 6 (the last figure mentioned by the reviewer) focus is therapeutic approaches to reducing RAN translation levels. While indeed GP levels are the only reported DPR levels, the stressors used in these experiments have already been shown to have robustly comparable DPR production in Figure 2.

Referee #2 (Remarks for Author):

The reviewers have addressed my major concerns. The following changes would be appropriate prior to publication.

Minor issues:

1) In introduction, the following reference should be added at the end of the first paragraph: "the production of polypeptides through the unconventional nonAUG-dependent translation of the NRE region, frequently referred to as repeat-associated nonAUG initiated (RAN) translation (Mori et al., 2013a; Zu et al., 2013; 2011; Ash et al, 2013).

We have included the additional reference suggested by the reviewer.

2) In introduction, the sentence below should be corrected as laid out to match existing nomenclature and to reference the correct work. "Recent mechanistic exploration of this translational phenomenon for the (G4C2)_n sense transcript has suggests that the non-AUG-dependent translation initiates predominantly from a near cognate AUG codon near cognate CUG codon upstream of the repeat (Kearse et al., 2016; Tabet et al., 2018; Todd et al., 2013 Green et al, 2017)"

We have included the additional reference suggested by the reviewer.

3) In introduction, the following sentences would be more correct if they were changed to the following: "One potential outcome driver of from these events is the dynamic translational regulation of mediated by phosphorylation of the α -subunit of eukaryotic translation initiation factor 2 (p-eif2 α). Peif2 α is phosphorylated in response to cellular stress, which suppresses global protein translational initiation. However, this same phosphorylation event favors non-canonical translation initiation events on mRNAs (Chesnokova et al., 2017), and is implicated to have a role in both CGG and C9orf72 NRE-linked RAN translation (Cheng et al., 2018; Green et al., 2017)

The text now includes the additional references suggested by the reviewer.

4) In results, change sentence "Given robustly roughly comparable RAN translation levels for all three frames of the C9 reporter construct in the presence of stressors, this and subsequent experiments utilized the GP frame C9 reporter which provides diffuse, uniform fluorescent signal for quantification."

We have updated the text as requested to say "*Since RAN translation in all three ORFs of the C9 reporter construct showed similar relative changes in levels in response to treatments, in the subsequent experiments we utilized only the GP frame, which provides diffuse, uniform fluorescent signal, to assess modulators of RAN translation.*"

Referee #3 (Remarks for Author):

The authors partially improved the manuscript by responding some issues raised by the reviewer;

however, many issues still remained unaddressed.

1. It has been known that many of poly dipeptides are easily smeared in PAGE gel except for poly (GA). For instances, Lee et al., demonstrated poly (GA) in WB (Hum Mol Gene, 2017). Zhang et al., also presented WB data of poly (GA) (Nat Neurosci, 2016). In this manuscript, the authors used artificial DRPs fused to Dendra2-HA; therefore, it is necessary to show the data of WB no matter how they look like.

We disagree with the reviewer, providing WB data no matter how they look like is not ideal for publication or assessment of relative DPR levels as reported in this work. The filter trap assay is used in the majority of publications where DPR relative levels of DPRs are being measured. Similar to others, we have applied this to our model system because it provides a focused protein density for detecting proteins of extremely low-abundance, which when separated potentially throughout the full length SDS-PAGE gel would be difficult to quantify. Additionally, filter trap data are rigorously validated and in agreement with the provided imaging data and quantitative analyses. This demonstrates the robustness of filter trap assays to measure relative levels and changes in DPR levels during screening with our DPR model system as reported in this work.

2. The authors are in a position to address frameshifting by comparing the expression levels of translated products by WB.

These analyses are very difficult to perform using our constructs and DPR expression levels in our tissue culture models, utilizing a reporter construct specifically designed to test frameshifting will greatly enhance the quantitative analyses of frameshifting and will provide valuable frequency, rates, and any frame preferences. A rigorous mechanistic investigation into frameshifting during translational elongation is necessary, and depending on the extent that frameshifting is occurring should be carefully measured due to its implications in numerous studies that have examined DPR pathology using DPR-specific antibodies and/or colocalization of DPRs. This investigation should be a major collaborative study that is beyond the scope of this work. Here, we demonstrate that all three ORFs are being produced based on DPR-specific antibodies regardless of the frame of the c-terminal Dendra2-HA fusion tag reporter. Therefore, if frame shifting is occurring somewhere within the repeats, we could have some contribution of the reporter Dendra2-HA that is fused to a hybrid DPR. However, our results for localization and half-lives based on the DPR-Dendra2-HA reporter, suggest that these potential hybrid DPR-Dendra2-HA have minimal contributions in our model systems used here. Previous work examining frameshifting within other repeat expansions have demonstrated that frameshifting is akin to a stochastic event that increases with the number of repeats (Girstmair, H. et al 2013 Cell Rep, PMID: 23352662). Further investigation into the extent and rates that frameshifting occurs during translational elongation will be performed in future experiments with tools specifically designed to quantitatively address such questions.

4. Fig 1C remains quite unreliable.

We believe the figure appropriately conveys the message that DPRs are localized in different compartments as expected based on numerous *in vitro* overexpression experiments in previous publication. Overexposure increases the visibility of the DPRs over the background of the iMFP fluorescence. The preponderant fluorescence of the iMFP signal is dictated by the prevalent canonical translation over non-AUG dependent translation, which is magnitudes less efficient. We have now included a reduced exposure image as a source data file to accompany this figure.

10. The data of Fig 5A must be shown in WB with size markers instead of filter trap assays as mentioned in the first round review comments. This was also requested by the other reviewer. Figure EV7C showed that there was almost no signals of P-Perk in the control lane which was not consistent with that in the filter trap assay in Fig 5.

The WB quantitative analysis closely matches the analysis with the filter trap assays. The filter trap blot picks up some background staining, which is shared in all the analyses. This could be the primary reason why there is more signal in the filter trap control lane. In addition, the exposure time of the full western blot is different from the filter trap blot. So the two signals are not directly comparable.

Corresponding Author Name: Aaron Haeusler, Davide Trotti

Manuscript Number: EMM-2018-09423